# Revisiting Mode Connectivity in Neural Networks with Bezier Surface

**Jie Ren**[1,2]**, Pin-Yu Chen**[3]**, Ren Wang**[1]*

[1]Illinois Institute of Technology, [2]University of Wisconsin-Madison, [3]IBM Research

## Abstract

Understanding the loss landscapes of neural networks (NNs) is critical for optimizing model performance. Previous research has identified the phenomenon of mode connectivity on curves, where two well-trained NNs can be connected by a continuous path in parameter space where the path maintains nearly constant loss. In this work, we extend the concept of mode connectivity to explore connectivity on surfaces, significantly broadening its applicability and unlocking new opportunities. While initial attempts to connect models via linear surfaces in parameter space were unsuccessful, we propose a novel optimization technique that consistently discovers Bézier surfaces with low-loss and high-accuracy connecting multiple NNs in a nonlinear manner. We further demonstrate that even without optimization, mode connectivity exists in certain cases of Bézier surfaces, where the models are carefully selected and combined linearly. This approach provides a deeper and more comprehensive understanding of the loss landscape and offers a novel way to identify models with enhanced performance for model averaging and output ensembling. We demonstrate the effectiveness of our method on CIFAR-10, CIFAR-100, and Tiny-ImageNet datasets using VGG16, ResNet18, and ViT architectures. The codes are available at https://github.com/TIML-Group/MCSurface.

## 1 Introduction

Understanding the loss landscapes of deep neural networks (DNNs) is crucial for understanding the training process in deep learning (Keskar et al., 2017). One of the most compelling phenomena observed within these landscapes is mode connectivity - trained models (modes) in a model's parameter space can be connected by paths of low loss (Garipov et al., 2018; Draxler et al., 2018). This phenomenon reveals that neural networks, even when initialized differently and converging to distinct optima, may share underlying structural relationships that allow for smooth transitions between them in parameter space. Understanding how these optima are connected can reveal key information about a network's performance across different tasks and datasets (Wortsman et al., 2022), as well as its stability and susceptibility to adversarial attacks (Zhao et al., 2020; Wang et al., 2024; 2023). One widely studied form of mode connectivity is linear mode connectivity, which describes a more specific scenario where the linear interpolation between two trained neural networks maintains a roughly constant loss (Frankle et al., 2020). Importantly, the discovery of mode connectivity has led to many applications such as exploring the safety alignment of large language models (Peng et al., 2024), where visualizing the safety landscape reveals how finetuning can compromise safety, and improved model robustness by finding paths in flat regions of the loss landscape (Zhao et al., 2020; Wang et al., 2024; Tatro et al., 2020), making models less sensitive to adversarial examples. Additionally, it enhances optimization efficiency by aiding escape from local minima and supports class incremental learning by enabling smooth connections between tasks without forgetting (Wortsman et al., 2022; Wen et al., 2023).

Despite the insights gained from mode connectivity, most existing studies focus on curve-based approaches that are inherently limited. These methods rely on information derived from just two endpoints—two neural networks with fixed parameters—restricting the exploration to a one-dimensional path. This limitation prevents the capture of richer structural information that exists

---

*Corresponding Author (Ren Wang: rwang74@iit.edu)

beyond these endpoints and confines the search space to a narrow region within the parameter space. The core motivation of this work is to extend mode connectivity from curves to surfaces, thereby enabling a broader and deeper exploration of the parameter space. By constructing surfaces instead of simple curves, we can integrate information from multiple models simultaneously, allowing for a more comprehensive search for optimal solutions. Surface connectivity offers a higher-dimensional perspective on the loss landscape, facilitating the discovery of models that exhibit better generalization and performance. In this paper, we present a novel learning framework that leverages Bézier surfaces in a nonlinear manner to explore mode connectivity based on Bézier surfaces (See Figure 1 for the loss landscape of Bézier surfaces before and after applying our method), offering deeper insights into neural network loss landscapes and opening up new avenues in model optimization. Additionally, We investigate the conditions under which the surface spanned by linear combinations of models exhibits mode connectivity. Finally, we introduce practical applications of our Bézier surface-based mode connectivity approach, including model averaging and output ensembling, demonstrating its potential to enhance performance in these tasks.

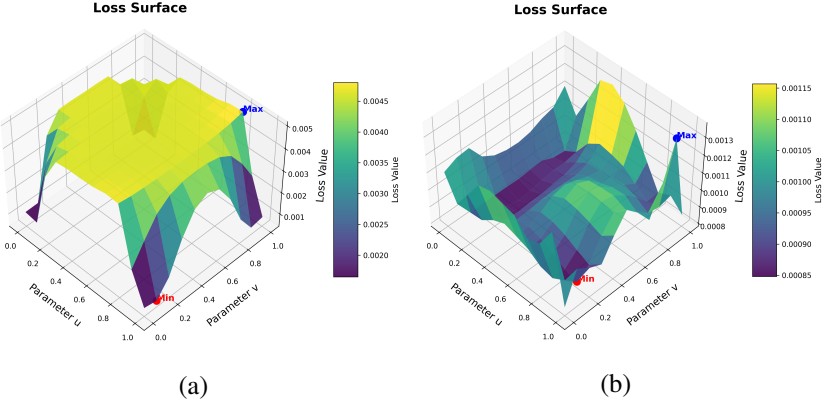

Figure 1: (a) The loss values on the Bézier surface are high before applying our method; (b) The loss values on the Bézier surface are consistently low after applying our method. Each point on the surface represents the loss obtained by a model. The Bézier surface is based on two parametric directions.

**Contributions**    This paper makes the following key contributions:

- **Bézier Surface-Based Mode Connectivity Framework and Algorithm:** This paper introduces a novel framework that extends mode connectivity from curves to surfaces using nonlinear Bézier surfaces. It also presents an efficient algorithm that constructs these Bézier surfaces to connect independently trained models, systematically optimizing the surface to maintain low loss and high accuracy across the entire parameter space. This approach provides a more comprehensive exploration of the neural network loss landscape.

- **Investigation of Linear Surface Connectivity:** The paper examines when surfaces formed by linear combinations of neural network models exhibit mode connectivity properties, identifying key conditions for this to occur. This provides a foundation for understanding the limits of linear surface-based approaches.

- **Applications of Bézier Surface Mode Connectivity:** (1) Model Averaging: The Bézier surface framework provides a new view for averaging models across the parameter space, leading to improved model performance; (2) Output Ensembling: The approach enhances model ensembling by connecting multiple models via Bézier surfaces, creating a more effective ensemble of models.

## 2    RELATED WORK

**Neural Network Loss Landscape and Mode Connectivity.**    Understanding the loss landscapes of deep neural networks is critical for analyzing their optimization and generalization behaviors. Despite the non-convex nature of neural network loss functions, overparameterized networks often converge to solutions with similar performance across different training runs. This observation has prompted investigations into the geometry of these loss landscapes, with a particular focus

on the sharpness and flatness of minima. Models trained with large batch sizes tend to converge to sharper minima, associated with poorer generalization compared to flatter minima found using smaller batches, emphasizing the relationship between the geometry of the minima and generalization abilities (Keskar et al., 2017). Mode connectivity explores the pathways in the loss landscape that connect different minima. Studies have shown that simple, low-loss paths - often through weight space interpolations - can connect independently trained models, indicating that minima are not isolated but reside within a connected manifold of low-loss regions (Garipov et al., 2018; Draxler et al., 2018). Techniques like Stochastic Weight Averaging (SWA) build on this concept by averaging weights along the trajectory of stochastic gradient descent (SGD) to find flatter minima that generalize better (Izmailov et al., 2018). Furthermore, regularization techniques such as dropout and batch normalization have been shown to promote smoother landscapes, which facilitate mode connectivity (Fort & Jastrzebski, 2019). Mode connectivity has also been applied in various tasks, including transfer learning (Wortsman et al., 2022), robustness (Wang et al., 2024), safety alignment of large language models (Peng et al., 2024), and class incremental learning (CIL) (Wen et al., 2023). For instance, it has been explored to study and enhance $\ell_p$ robustness and to fine-tune pre-trained models for improved performance on downstream tasks (Wortsman et al., 2022; Zhao et al., 2020; Wang et al., 2024; 2023). Additionally, mode connectivity has been utilized as a post-processing step after each CIL phase to optimize continual learning performance (Wen et al., 2023). Several previous works explored the extensions of mode connectivity. Some focused on identifying specific patterns within loss surfaces of multiple neural networks (Skorokhodov & Burtsev, 2019; Czarnecki et al., 2019). Others explored low-loss spaces using multiple simplices (Benton et al., 2021), which rely on localized, piecewise-linear approximations, or modeled the loss landscape as a collection of high-dimensional wedges (Fort et al., 2019), which formulate the problem as a linear interpolation of manifolds and face challenges in scalability and interpretability. In contrast, our method focuses on nonlinear, Bézier surface-based mode connectivity, offering a global and smooth mapping, enabling efficient optimization, and enhancing visualization. While one study examined geometry across multiple loss subspaces, its focus was on pairwise mode connectivity and did not extend to constructing surfaces with provable low-loss properties (Chen & Saidi). Despite these advancements, most works focus on curve-based approaches, which restrict exploration to narrow regions of the parameter space. Our method uniquely emphasizes the exploration of mode connectivity through surfaces, providing insights beyond the scope of these prior works.

**Linear Mode Connectivity.** While mode connectivity typically involves non-linear paths between minima, an intriguing possibility is the existence of linear paths that connect different minima without encountering high-loss barriers. Linear Mode Connectivity (LMC) refers to a special case of mode connectivity where the low-loss path between two trained models is a linear interpolation (Frankle et al., 2020), which exists when models share initial weights or training data ordering. Further theoretical work has demonstrated that overparameterization in neural networks makes linear low-loss paths more common (Kuditipudi et al., 2019), and techniques such as weight permutation alignment allow for LMC between independently trained models by reordering neurons to match across models (Entezari et al., 2022). These findings have practical implications for model merging and model ensembling, where linear combinations of weights can result in averaging or ensembles that outperform traditional methods (Rame et al., 2022; Ainsworth et al., 2023). While these works primarily focus on linear combinations between two models, we extend the exploration to settings involving more than two models, generalizing the idea of linear mode connectivity.

## 3 BÉZIER CURVE AND BÉZIER SURFACE

**Bézier Curve.** A Bézier curve is defined using a parametric equation where the parameter $t$ varies between 0 and 1. The

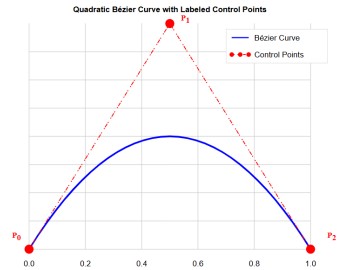

(a) Bézier curve

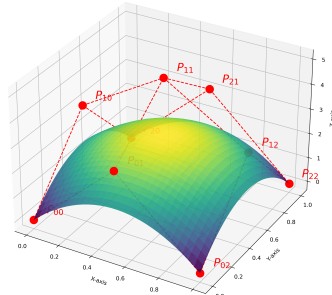

(b) Bézier surface

Figure 2: An illustration of Bézier curve and Bézier surface. The Bézier surface is based on two parametric directions $u \in [0, 1]$ and $v \in [0, 1]$, with control points arranged in a grid.

general form of a Bézier curve of degree $n$ is given by:

$$B(t) = \sum_{i=0}^{n} \binom{n}{i} t^i (1-t)^{n-i} P_i, \quad \binom{n}{i} = \frac{n!}{i!(n-i)!}, \tag{1}$$

where $P_i$ represent the control points. When $n = 2$, equation (1) reduces to the most commonly used quadratic Bézier curve with three control points ($P_0$, $P_1$, and $P_2$):

$$B(t) = (1-t)^2 P_0 + 2t(1-t)P_1 + t^2 P_2, \tag{2}$$

which is illustrated in Figure 2 (a). In the context of mode connectivity, $P_i$ are neural networks with the same architecture, and $B(t)$ denotes a curve in the parameter space.

**Bézier Surface.** A Bézier surface is an extension of the Bézier curve from $t$ to two parametric directions, $u \in [0, 1]$ and $v \in [0, 1]$, with control points arranged in a grid. The surface is defined as:

$$B(u, v) = \sum_{i=0}^{n} \sum_{j=0}^{m} P_{ij} B_{i,n}(u) B_{j,m}(v), \tag{3}$$

where $P_{i,j}$ are control points, and $B_{i,n}(u)$ and $B_{j,m}(v)$ are the Bernstein polynomial:

$$B_{i,n}(u) = \binom{n}{i} u^i (1-u)^{n-i}, \quad B_{j,m}(v) = \binom{m}{j} v^j (1-v)^{m-j} \tag{4}$$

The Bézier surface can be expanded by first summing over $j$ and then over $i$, yielding:

$$B(u, v) = \sum_{j=0}^{m} B_{j,m}(v) \left[ \sum_{i=0}^{n} B_{i,n}(u) P_{ij} \right] \tag{5}$$

When $u$ and $v$ equal to 0 or 1, we get the four corner control points that always lie on the Bézier surface: $B(0,0) = P_{00}, B(0,1) = P_{0m}, B(1,0) = P_{n0}, B(1,1) = P_{nm}$. The Bézier surface is illustrated in Figure 2 (b). Here, we use 9 control points by setting $n = m = 2$. Four of them are placed at the corners. The remaining five points do not lie directly on the surface (see the red points outside of the surface). Together, these nine models generate the entire parameter surface by varying $u$ and $v$. Similar to the Bézier curve case, $P_{ij}$ are neural networks in the context of mode connectivity (We use $\boldsymbol{\theta}_{ij}$ to represent neural networks in the following sections), with $B(u, v)$ denoting a low-loss surface in the parameter space. In this paper, our focus is on the surface scenario.

## 4   MODE CONNECTIVITY ON SURFACES

Exploring mode connectivity through curves has already yielded valuable insights into the structure of the loss landscape. However, examining the landscape using surfaces provides even deeper insights, offering a more comprehensive understanding of how models traverse these spaces. Unlike curves, surfaces enable the exploration of a broader parameter space and incorporate information from more models, which can lead to the development of more advanced techniques applicable across different domains. In this section, we outline our approach to constructing them and discuss the advantages of using surfaces for mode connectivity. The central question we seek to answer is:

> *How can we find a surface that encompasses models with both low training loss and high test accuracy?*

### 4.1   A PILOT EXPLORATION IN LINEAR SETTINGS

To answer this question, we begin with a pilot exploration in the simplest case, where the surface in the parameter space is spanned by a linear combination of three models. Consider these three models, $\boldsymbol{\theta}_1$, $\boldsymbol{\theta}_2$, and $\boldsymbol{\theta}_3$, where the coefficients satisfy $\alpha_1 + \alpha_2 + \alpha_3 = 1$ and $\alpha_i$ can vary. These coefficients form a surface parameterized by:

$$\phi(\alpha_1, \alpha_2, \alpha_3) = \alpha_1 \cdot \boldsymbol{\theta}_1 + \alpha_2 \cdot \boldsymbol{\theta}_2 + \alpha_3 \cdot \boldsymbol{\theta}_3$$

One can see that given a fixed group of $(\alpha_1, \alpha_2, \alpha_3)$, $\phi$ is a fixed model on the surface. Therefore, there is no optimization or learnable variables involved in the surface generation.

In our experiments, we train these three models independently with different initializations. This experiment is conducted on the CIFAR10 dataset using VGG16 architecture. By varying $\alpha_1, \alpha_2,$ and $\alpha_3$, we generate a surface through the linear interpolation of these three model parameters. We then evaluate the accuracy of the models on this surface and present the results in Figure 3.

From Figure 3, it is evident that the surface generated by linear combinations of models does not achieve low loss and high accuracy. In most cases, linear combinations of different models fail to create a surface with these desirable properties. This raises the question: if simple linear combinations are inadequate, can we learn such a surface through nonlinear methods?

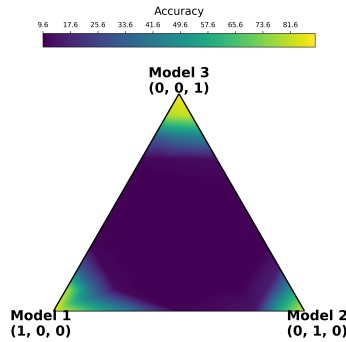

Figure 3: Each point on the triangular surface represents the accuracy of a model obtained through a linear combination of the three corner models in parameter space. The accuracies are low on the parameter surface, which is defined by the simple linear combination of three distinct models trained with different initializations in the parameter space.

## 4.2 Mode Connectivity on a Bézier Surface

Here, we propose a novel optimization method that addresses the surface construction question in a nonlinear manner. Our approach leverages Bézier surfaces in equation (3) to connect multiple models, consistently yielding a smooth surface with low loss and high accuracy.

**Connecting Procedure.** We begin by explaining how the models are connected on the given surface[1]. We redefine equation (3) from the perspective of model combinations:

$$\phi_{\boldsymbol{\theta}}(u, v) = \sum_{i=0}^{n} \sum_{j=0}^{m} B_{i,n}(u) B_{j,m}(v) \boldsymbol{\theta}_{ij}, \tag{6}$$

where $\boldsymbol{\theta} = \{\boldsymbol{\theta}_{ij}\}, i \in \{0, 1, 2, \cdots, n-1\}, j \in \{0, 1, 2, \cdots, m-1\}$ are the learnable control points, except for $\boldsymbol{\theta}_{00}, \boldsymbol{\theta}_{0m}, \boldsymbol{\theta}_{n0},$ and $\boldsymbol{\theta}_{nm}$, which are weights of four independently trained neural networks. These four control points have fixed parameters and will always be on the surface. We will use the four control points as anchors of the surface, and the remaining control points are free to be optimized.

**Learning Objective.** Next, we define the objective of the proposed surface optimization problem. The core idea is to learn a Bézier surface where all points exhibit low-loss and high-accuracy properties. Notice that the Bézier surface is defined by control points, with the four corner control points fixed, as they already demonstrate acceptable performance. By optimizing the remaining trainable control points $\boldsymbol{\theta}$, we can enhance the performance across the surface. We introduce our loss $\hat{l}(\theta)$ on the entire surface $\phi_{\boldsymbol{\theta}}(u, v)$ over uniform distribution below:

$$\hat{\ell}(\boldsymbol{\theta}) = \int_0^1 \int_0^1 L(\phi_{\boldsymbol{\theta}}(u, v)) q_{uv}(u, v)\, du\, dv = \mathbb{E}_{(u,v) \sim q_{uv}(u,v)} \left[ L(\phi_{\boldsymbol{\theta}}(u, v)) \right], \tag{7}$$

Here, we specify the loss function $L$ as task specific loss on the surface, we use cross-entropy loss to calculate different between the output of the network and the target classes for our following description. And distribution $q_{uv}(u, v)$ on $(u, v) \in [0, 1]^2$ is defined as:

$$q_{uv}(u, v) = \left\| \frac{\partial \phi_{\boldsymbol{\theta}}}{\partial u} \times \frac{\partial \phi_{\boldsymbol{\theta}}}{\partial v} \right\| \cdot \left( \int_0^1 \int_0^1 \left\| \frac{\partial \phi_{\boldsymbol{\theta}}}{\partial u} \times \frac{\partial \phi_{\boldsymbol{\theta}}}{\partial v} \right\| du\, dv \right)^{-1}, \tag{8}$$

where $q_{uv}$ represents the normalized density of points on the Bézier surface, weighted by the gradient of the parameterized surface. This density accounts for the varying distribution of points across the surface, ensuring the loss integral reflects the true geometric properties of the surface. Direct

---

[1]We remark that our method can be easily generalized to more types of nonlinear surfaces. We choose Bézier surface because of its simplicity.

computation of $q_{uv}$ is intractable for stochastic gradient-based optimization because it relies on the gradients of the parameterization $\phi_{\boldsymbol{\theta}}(u, v)$, where $\phi_{\boldsymbol{\theta}}(u, v)$ depends on learned parameters $\boldsymbol{\theta}$.

Then the numerator of $\hat{\ell}(\boldsymbol{\theta})$ is the surface integral of the loss $L(\phi_{\boldsymbol{\theta}})$ on the surface, and the denominator is the normalizing constant of the uniform distribution on the surface defined by $\phi_{\boldsymbol{\theta}}(\cdot, \cdot)$.

We aim to minimize this loss with respect to $\boldsymbol{\theta}$. However, due to the computational intractability of directly optimizing this loss, we propose a more computationally feasible loss below:

$$\ell(\boldsymbol{\theta}) = \int_0^1 \int_0^1 L(\phi_{\boldsymbol{\theta}}(u, v)) du \, dv = \mathbb{E}_{(u,v) \sim U([0,1]^2)} \left[ L(\phi_{\boldsymbol{\theta}}(u, v)) \right] \tag{9}$$

where $U([0, 1]^2)$ represents the uniform distribution over $[0, 1]^2$. This yields an expectation of $L(\phi_{\boldsymbol{\theta}}(u, v))$ with respect to a uniform distribution on $(u, v) \in [0, 1]^2$, while $\hat{\ell}(\boldsymbol{\theta})$ is an expectation with respect to a uniform distribution on the surface. The two losses align, for instance, when $\phi_{\boldsymbol{\theta}}(\cdot, \cdot)$ defines a bilinear surface with four corner control points and linear parametrization in $u$ and $v$. To minimize $\ell(\boldsymbol{\theta})$, the intuitive way is to sample $\tilde{u}, \tilde{v}$ from the uniform distribution $U([0, 1]^2)$ and make a gradient step for $\boldsymbol{\theta}$ with respect to the loss $L(\phi_{\boldsymbol{\theta}}(\tilde{u}, \tilde{v}))$. This way, we obtain unbiased estimates of the gradients of $\ell(\boldsymbol{\theta})$, as

$$\nabla_{\boldsymbol{\theta}} L(\phi_{\boldsymbol{\theta}}(\tilde{u}, \tilde{v})) \overset{\text{def}}{=} \mathbb{E}_{(u,v) \sim U([0,1]^2)} \left[ \nabla_{\boldsymbol{\theta}} L(\phi_{\boldsymbol{\theta}}(u, v)) \right] = \nabla_{\boldsymbol{\theta}} \ell(\boldsymbol{\theta}). \tag{10}$$

We repeat these updates until convergence. To achieve this in a more efficient way, we employ a three-phase optimization process in the Learning Algorithm below.

**Learning Algorithm.** Minimizing equation (9) necessitates updating all control models simultaneously, which complicates the training process and is inefficient if done from scratch. Therefore, we found that ensuring the edge curves formed by control points achieve low loss and high accuracy can help shape the surface, resulting in the entire surface that also exhibits low loss and high accuracy properties. To improve efficiency, we adopt an "**outer-to-inner**" optimization strategy, progressing from curves to the surface. The idea is that optimizing the surface becomes significantly easier once the key curves achieve acceptable performance. Therefore, we first focus on optimizing the two parallel curves at the surface edges when $u = 0$ and $1$ while $v$ changes. (**Phase 1**), and then update the entire surface by refining all the trainable control points (**Phase 2**). The explanation of each phase is provided below, with the algorithm outlined in Algorithm 1. We also provide a detailed algorithm in the Appendix.

**Phase 1**: In this phase, we focus on optimizing the control points along the horizontal curves at the surface edges, parameterized by $v$ when $u = 0$ and $u = 1$. This optimization is performed iteratively over $E_1$ epochs. As seen in equation (5), only the control points $\boldsymbol{\theta}_{0j}$ and $\boldsymbol{\theta}_{nj}$ ($j \in \{1, \ldots, m - 1\}$) contribute to the updates when $u = 0$ and $u = 1$, while the corner points remain fixed. Phase 1 corresponds to lines 2-4 in Algorithm 1.

**Phase 2**: In this phase, we optimize all control points for $E_2$ epochs, except the fixed corner control points, by uniformly sampling over $u$ and $v$ and minimizing the expected loss across the entire surface. This process is repeated until convergence, producing a smooth Bézier surface that connects the corner points while minimizing the loss. Phase 2 corresponds to lines 5-7 in Algorithm 1.

To demonstrate the effectiveness of our method, we present results using the VGG16 architecture on the CIFAR-10 dataset when $n = m = 2$ (with 9 control points for the surface). Figures 4 and 5 illustrate the loss and accuracy surfaces before and after training, respectively. Unlike Figure 1, here we show loss and accuracy values on the surface in 2D. Once the optimization is complete, all $\boldsymbol{\theta}$ values are fixed, spanning the surface along $u$ and $v$. Each point in the figures represents a loss or accuracy value obtained by a model on the surface, evaluated on the test data. As shown, our method successfully optimizes the surfaces, transforming them from high loss (Figure 4 (a)) to low loss (Figure 4 (b)) and from low accuracy (Figure 5 (a)) to high accuracy (Figure 5 (b)). These results confirm that our method can effectively identify surfaces with low loss and high accuracy. One can also see from these two figures that the valleys in the training loss surface correspond closely to the peaks in test accuracy, enabling the efficient selection of optimal models without extensive test set evaluations. Further discussions and additional results across various datasets and model architectures are provided in the Experimental Section. We further propose a more efficient method by updating only specific layers of the model, significantly reducing computational overhead. Details of this layer-specific optimization approach can be found in the Appendix.

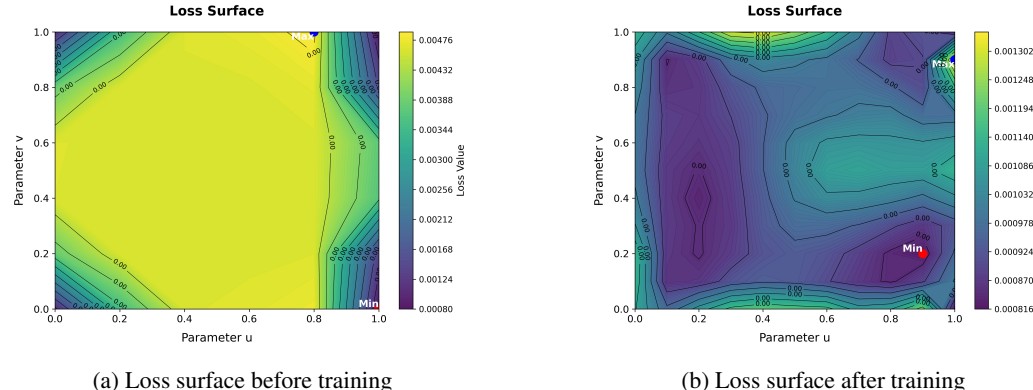

(a) Loss surface before training

(b) Loss surface after training

Figure 4: The losses on the Bézier surface decrease significantly after training. Each point represents a loss value evaluated on the test data. There are nine control points ($n = m = 2$). Loss values at $(u, v) = (0, 0), (0, 1), (1, 0), (1, 1)$ correspond to four corner control points $\boldsymbol{\theta}_{00}, \boldsymbol{\theta}_{20}, \boldsymbol{\theta}_{02}$, and $\boldsymbol{\theta}_{22}$.

---

**Algorithm 1:** Bezier Surface Mode Connectivity Algorithm (Summary)

**Input:** Initial weights of the four corner control points $\boldsymbol{\theta}_{00}, \boldsymbol{\theta}_{n0}, \boldsymbol{\theta}_{0m}$, and $\boldsymbol{\theta}_{nm}$, number of total epochs $E = E_1 + E_2$, number of random samples $k$, batch size $B$
**Output:** Optimized Bezier surface control points

1  Initialize control points $\boldsymbol{\theta}$

2  **for** *epoch from* $0$ *to* $E_1$ **do**
3     Sample $B$ training data in each batch and sample $u \sim U(0, 1)$ for $k$ times
4     Update curves $\phi_{\boldsymbol{\theta}}(0, v)$ and $\phi_{\boldsymbol{\theta}}(1, v)$ from sampled points by minimizing equation (9);

5  **for** *epoch from* $0$ *to* $E_2$ **do**
6     Sample $B$ training data in each batch and sample $u, v \sim U(0, 1)$ for $k$ times
7     Update full Bézier surface $\phi_{\boldsymbol{\theta}}(u, v)$ from sampled points by minimizing equation (9).

---

**Existence of Linear Surface Mode Connectivity.** While we have shown that linear combinations of models often fail to produce a desirable surface, we have identified special cases where they can succeed. Specifically, when $n = 4$ and any two pairs of these four models satisfy the linear mode connectivity property, it is possible to construct a Bézier surface where every model exhibits low

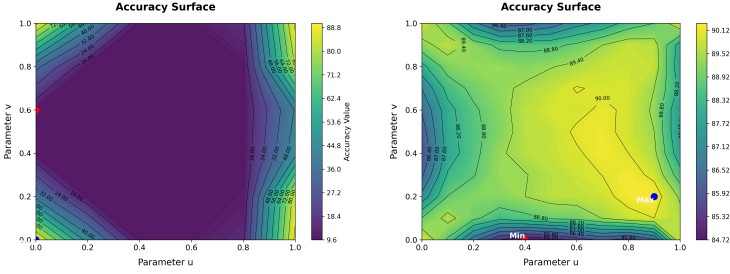

(a) Accuracy surface before training

(b) Accuracy surface after training

Figure 5: The accuracies on the Bézier surface increase significantly after training. Each point in the figures represents an accuracy value obtained by a model on the surface, evaluated on the test data. There are nine control points ($n = m = 2$). Accuracy values at $(u, v) = (0, 0), (0, 1), (1, 0), (1, 1)$ are the values obtained by four corner control points $\boldsymbol{\theta}_{00}, \boldsymbol{\theta}_{20}, \boldsymbol{\theta}_{02}$, and $\boldsymbol{\theta}_{22}$.

loss and high accuracy. By selecting $\boldsymbol{\theta}_{00}, \boldsymbol{\theta}_{n0}, \boldsymbol{\theta}_{0m}$, and $\boldsymbol{\theta}_{nm}$ from a training trajectory, we ensure that the pairs of these models meet the linear mode connectivity condition. The interior control points $\boldsymbol{\theta}_{ij}$ are then obtained via linear interpolation of the four corner points, with $\boldsymbol{\theta}_{ij}$ calculated as a weighted combination - $\boldsymbol{\theta}_{ij} = (1 - t)(1 - s)\boldsymbol{\theta}_{00} + t(1 - s)\boldsymbol{\theta}_{n0} + (1 - t)s\boldsymbol{\theta}_{0m} + ts\boldsymbol{\theta}_{nm}$, where $t = \frac{i}{n}$ and $s = \frac{j}{m}$. The models on the surface are thus pure linear combinations of the four initial models. We demonstrate that this surface exhibits low loss and high accuracy using VGG16 on the CIFAR-10 dataset, as shown in Figure 6. We select the four corner points from epochs 220,

200, 180, and 160, then combine them using the process outlined above. As a result, the surface demonstrates similarly low loss and high accuracy, comparable to that of the four corner points.

### 4.3 Applications of Mode Connectivity with Bézier Surface

While this paper primarily focuses on introducing a surface mode connectivity method to explore the neural network loss landscape, we also aim to demonstrate its potential for various applications. In this section, we explore two key applications of our proposed Bézier surface-based mode connectivity framework: model averaging and output ensembling. By extending mode connectivity from curves to surfaces, we enable more effective methods for model optimization and generalization.

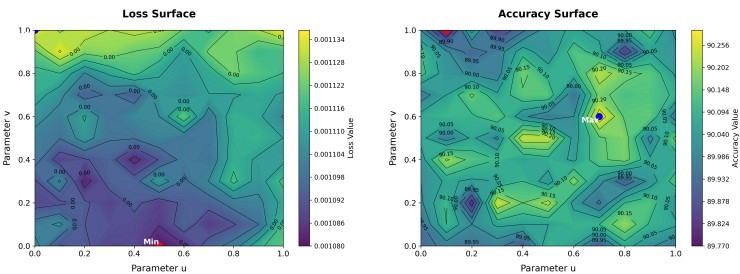

(a) Loss values on the surface      (b) Accuracy values on the surface

Figure 6: The Bézier surface linearly spanned by the four corner points shows similarly low loss and high accuracy, matching the performance of the corner points. The four corner points are VGG16 models selected from epochs 220, 200, 180, and 160 during the training on CIFAR-10 dataset.

**Model Merging.** The concept of model soups, first introduced by Wortsman et al. (Wortsman et al., 2022), involves averaging the weights of multiple fine-tuned models to improve accuracy without increasing inference time. Unlike traditional ensemble methods, which aggregate the outputs of multiple models, model soups merge the weights of independently fine-tuned models, leveraging the observation that models trained on the same dataset or task often reside in similar low-loss basins of the error landscape. This principle aligns with findings in linear mode connectivity, which suggests that multiple neural networks can be connected by paths of non-increasing loss between them (Garipov et al., 2018; Draxler et al., 2018).

Building on these ideas, our proposed Bézier surface connectivity framework acts as a model merging method that enables a more thorough exploration of the model parameter space by averaging across surfaces rather than linear paths. Unlike conventional model merging methods such as Lion et al. (2024), which operate under the constraint that models lie within a single basin, our method allows for connecting and merging models across basins. This enables a wider utilization of diverse models. Since each control point lies in the parameter space and represents a model, surface-based averaging allows us to connect more than two models, leading to the discovery of new, high-performing models that outperform those created by traditional weight averaging techniques. As seen in equation (6), each point on the surface represents a model merged with different weights. In our implementation, we select the point on the surface that yields the best performance (Model Merging Accuracy), and compare it with the average accuracy of the four corner control models (Avg Initial Model Accuracy). As shown in Table 1, we report the accuracy of the best-performing model on the surface. Our experimental results show that model merging through Bézier surfaces consistently outperforms the component models, achieving better generalization and accuracy. Additionally, in Figure 5 (b), you can see a highland on the accuracy surface, indicating regions with higher accuracy than the four initial corner models. We remark that model merging can also be conducted in our linear surface mode connectivity setting.

**Output Ensembling.** Ensembling techniques traditionally combine the predictions of multiple models to improve generalization. However, the use of simple linear averaging in these methods limits the potential to fully explore the parameter space. Recent advancements in mode connectivity research (Wortsman et al., 2021; Rame et al., 2022) have demonstrated that models can be connected by paths of constant or non-increasing loss in parameter space, which leads to more robust ensembling strategies. Our Bézier surface-based method expands upon these findings by connecting up to four independent models, allowing for a broader and more nuanced exploration of the parameter space. Unlike linear mode connectivity, which only considers paths between two models,

surface-based connectivity explores multidimensional relationships, resulting in models that generalize better on different tasks and datasets.

In our approach, we perform model ensembling by combining the output probabilities of different models positioned on the Bézier surface. To do this, we uniformly sample values of $u$ and $v$ at regular intervals (defaulting to 0.1 in our experiments) across the surface, creating a diverse set of models. By averaging the outputs of these sampled models, we generate ensemble predictions. We found that this ensembling strategy consistently led to improved accuracy, surpassing the performance of the best individual model on the Bézier surface.

# 5 EXPERIMENTS

## 5.1 EXPERIMENT SETTINGS

The main goal of this section is to further demonstrate that the methods proposed in Section 4 successfully identify low-loss, high-accuracy surfaces connecting given modes across various architectures and datasets. In particular, we evaluate our method on three different datasets including CIFAR-10 (Krizhevsky & Hinton, 2009), CIFAR-100 (Krizhevsky & Hinton, 2009), and Tiny-Imagenet (Le & Yang, 2015) using ResNet18 (He et al., 2016), VGG16 (Simonyan & Zisserman, 2015), and ViT model architectures. In this paper, we primarily use CIFAR-10 and VGG16 as the default dataset and architecture for presenting loss and accuracy surface plots. By default, we set $n = m = 2$ unless otherwise specified.

## 5.2 MODE CONNECTIVITY ON BÉZIER SURFACE

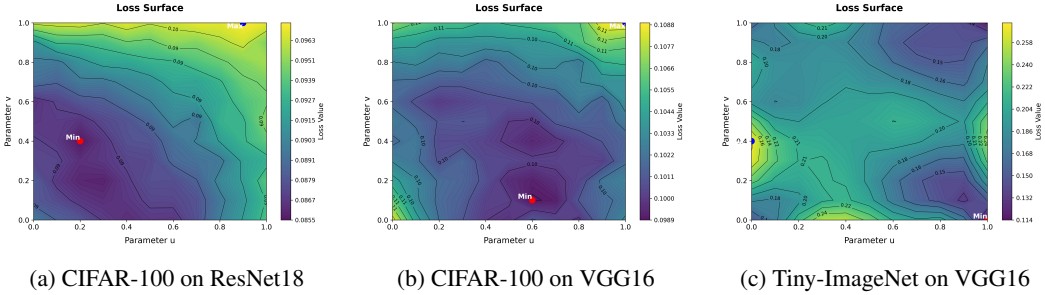

(a) CIFAR-100 on ResNet18   (b) CIFAR-100 on VGG16   (c) Tiny-ImageNet on VGG16

Figure 7: Our Bézier surface approach can effectively identify low-loss surfaces after training on various datasets and architectures. The loss values are similarly small across the surfaces.

**Results on More Datasets & Architectures.** We first evaluate the effectiveness of our Bézier surface-based mode connectivity method by examining the loss and accuracy surfaces after training on a variety of datasets and architectures. The surfaces depicted in Figure 7 illustrate our method's ability to identify low-loss landscapes, enhancing optimization in diverse settings. In Table 1, we present results on more diverse datasets and architectures, and show comparisons of Model Merging Accuracy, Average Surface Accuracy, and Average Corner Accuracy achieved by our method. The Avg Corner Accuracy refers to the average accuracy of the four corner control models, while the Avg Surface Accuracy represents the average accuracy of the sampled models on the surfaces. The Model Merging Accuracy denotes the highest accuracy found on the Bézier surface. The results demonstrate that our method consistently identifies low-loss surfaces (surface mode connectivity) across different settings. In the Appendix, we conducted additional experiments on surfaces with more control points. The results show that even with increased complexity, Bézier surfaces consistently maintain low training loss and high test accuracy properties across all configurations.

**Linear Surface Mode Connectivity.** We have shown the result of applying this approach to the VGG16 architecture on CIFAR-10 in Figure 6. Here, we repeat the experiment using ResNet18 on CIFAR-100. We select four corner points from epochs 220, 200, 180, and 160 during the training. Similarly, we found that the linear combinations of models create a surface with desirable properties,

Table 1: Comparisons of Model Merging Accuracy, Average Surface Accuracy, and Average Corner Accuracy obtained by our method across different datasets and architectures. The Avg Corner Accuracy represents the average accuracy of the four corner control models. The Avg Surface Accuracy is the average accuracy of the models on the surfaces (by sampling). The Model Merging Accuracy is the highest accuracy obtained on the Bézier surface. The results show that (1) our method can always find low-loss surfaces in different settings; (2) model merging through the Bézier surface can discover models with improved performance.

| DATASET | MODEL | Model Merging Accuracy | Avg Surface Accuracy | Avg Corner Accuracy |
|---|---|---|---|---|
| CIFAR-10 | ResNet18 | **90.9%** | 89.3% | 90.1% |
| CIFAR-10 | VGG16 | **90.7%** | 89.8% | 89.6% |
| CIFAR-10 | ViT | **70.2%** | 65.4% | 70.0% |
| CIFAR-100 | VGG16 | **70.8%** | 69.3% | 70.7% |
| Tiny-ImageNet | VGG16 | **51.1%** | 47.4% | 50.7% |

demonstrating high accuracy throughout, as shown in Figure 8(b). The loss surface is shown in the Appendix. Across both experiments, we observed that the central region of the surface typically exhibited slightly lower loss and higher accuracy compared to the corner models, further supporting the utility of linear surface mode connectivity in certain settings.

### 5.3 MODEL MERGING AND ENSEMBLING BASED ON SURFACE MODE CONNECTIVITY

We further explore model merging and output ensembling to highlight the potential applications of surface mode connectivity.

**Model Merging.** For model Merging, we obtain the optimal model on the surface to evaluate model merging. Table 1 demonstrates the boost in validation accuracy for models obtained from the surface using our Bezier surface as a model merging method compared to the four corner control points. We obtained the model with high accuracy. You may also check the validation accuracy surface after training shown in Figure 5, showing that there's a plateau with higher accuracy than four models on the surface. The central high accuracy region in Figure 8(b) also illustrates that linear surface mode connectivity can be utilized for model merging without the need for optimization.

**Output Ensembling.** For model ensembling, we sampled values of $u$ and $v$ uniformly between 0 and 1 at intervals of 0.1, generating diverse models along the Bézier surface. By averaging the output probabilities from these sampled models, we created an ensemble that consistently improved accuracy. In our experiments, we apply the Bézier surface-based model ensembling method on the CIFAR-10 dataset using VGG16 models, which results in notable accuracy improvements. Specifically, the average accuracy of the four independently trained models (corner points) was 89.96%, while the best-performing model on the Bézier surface achieved 90.70%. After applying our ensembling technique, the accuracy further increased to 92%, demonstrating the effectiveness of surface-based model ensembling in enhancing generalization and outperforming both individual models and traditional linear interpolation methods.

## 6 CONCLUSION

In this paper, we extend the concept of mode connectivity from curves to surfaces, offering a broader exploration of neural network loss landscapes. By leveraging Bézier surfaces, we demonstrate the ability to connect multiple models with low-loss and high-accuracy regions, enhancing both model merging and optimization. Our approach not only uncovers deeper insights into the structure of the parameter space but also improves model averaging and ensembling performance. We validate the effectiveness of this method across various datasets and architectures, including CIFAR-10, CIFAR-100, and Tiny-ImageNet.

## 7 ACKNOWLEDGEMENT

This work is supported by the NSF under Grants 2246157 and 2319243. We are thankful for the computational resources made available through NSF ACCESS and Argonne Leadership Computing Facility.

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

# A APPENDIX

## A.1 ALGORITHM DETAILS

We enclose the full algorithm below

---

**Algorithm 2:** Bezier Surface Mode Connectivity Algorithm

---

**Input:** Initial weights $\boldsymbol{\theta}_{00}, \boldsymbol{\theta}_{nm}, \boldsymbol{\theta}_{0m}$, and $\boldsymbol{\theta}_{n0}$ (fixed four end control points), number of epochs $E_1, E_2$, with epoch $E = E_1 + E_2$, learning rate $\eta$, number of random samples $k$, training dataset $D_0$, batch size $B$

**Output:** Optimized Bezier surface control points

---

1  Initialize control points $\boldsymbol{\theta}_{ij}$ with linear interpolation
2  **for** $i, j \in \{0, \ldots, n\}$ **do**
3      **if** $(i, j) \neq (0, 0), (0, m), (n, 0), (n, m)$ **then**
4          Let $t = \frac{i}{n}$ and $s = \frac{j}{m}$
5          Initialize $\boldsymbol{\theta}_{ij}$ using linear interpolation as:
6          $\boldsymbol{\theta}_{ij} = (1-t)(1-s)\boldsymbol{\theta}_{00} + t(1-s)\boldsymbol{\theta}_{n0} + (1-t)s\boldsymbol{\theta}_{0m} + ts\boldsymbol{\theta}_{nm}$

7  Define $\boldsymbol{\theta}_{00}, \boldsymbol{\theta}_{0m}, \boldsymbol{\theta}_{n0}$, and $\boldsymbol{\theta}_{nm}$ as fixed endpoints
8  **for** *epoch e in 1 to $E_1$* **do**
9      **for** *each data batch $D_b \in D_0$* **do**
10         **for** *v in sampled $v_i \sim U(0,1)$* **do**
11             $B_1(0, v) = \boldsymbol{\theta}_{00}B_{0,m}(v) + \boldsymbol{\theta}_{01}B_{1,m}(v) + \cdots + \boldsymbol{\theta}_{0m}B_{m,m}(v)$
12             $B_2(1, v) = \boldsymbol{\theta}_{n0}B_{0,m}(v) + \boldsymbol{\theta}_{n1}B_{1,m}(v) + \cdots + \boldsymbol{\theta}_{nm}B_{m,m}(v)$
13             **for** *each $x \in D_b$* **do**
14                 compute loss $l(\boldsymbol{\theta}) = \frac{1}{k}\sum_{v_i}\left(L_{task}(B_1(0, v_i); x) + L_{task}(B_2(1, v_i); x)\right)$
15                 Compute gradients $\nabla\boldsymbol{\theta}_{ij} = \frac{\partial l}{\partial \boldsymbol{\theta}_{ij}}$ for each learnable $\boldsymbol{\theta}_{ij}, i \in \{0, n\}$
16                 Update $\boldsymbol{\theta}_{ij} \leftarrow \boldsymbol{\theta}_{ij} - \eta\nabla\boldsymbol{\theta}_{ij}$

17 **for** *epoch e in $E_1 + 1$ to $E_1 + E_2$* **do**
18     **for** *each data batch $D_b \in D_0$* **do**
19         **for** *v in sampled $v_i \sim U(0,1)$ and u in sampled $u_i \sim Uniform(0,1)$* **do**
20             $B(u, v) = \sum_{i=0}^{n}\sum_{j=0}^{m}\boldsymbol{\theta}_{ij}B_{i,n}(u)B_{j,m}(v)$
21         **for** *each $x \in D_b$* **do**
22              compute loss $L = \frac{1}{k}\sum_{u_i, v_i} L_{task}(B(u_i, v_i))$
23              Compute gradients $\nabla\boldsymbol{\theta}_{ij} = \frac{\partial L}{\partial \boldsymbol{\theta}_{ij}}$ for each learnable $\boldsymbol{\theta}_{ij}$
24              Update $\boldsymbol{\theta}_{ij} \leftarrow \boldsymbol{\theta}_{ij} - \eta\nabla\boldsymbol{\theta}_{ij}$

25     empty $sampled\_points$

---

## A.2 LINEAR SURFACE MODE CONNECTIVITY

We construct a Bézier surface where the control points are the learnable parameters, denoted as $\boldsymbol{\theta_{ij}}$. The intermediate control points on the surface are calculated as linear combinations of four corner models $\boldsymbol{\theta_{00}}, \boldsymbol{\theta_{n0}}, \boldsymbol{\theta_{0m}}$, and $\boldsymbol{\theta_{nm}}$, as follows:

$$\boldsymbol{\theta_{ij}} = (1-t)(1-s)\boldsymbol{\theta_{00}} + t(1-s)\boldsymbol{\theta_{n0}} + (1-t)s\boldsymbol{\theta_{0m}} + ts\boldsymbol{\theta_{nm}}$$

where $t = \frac{i}{n}$ and $s = \frac{j}{m}$. This results in a surface composed of linear combinations of the four initial models.

We first applied this approach with the ResNet18 architecture on the CIFAR-100 dataset, selecting corner points from epochs 220, 200, 180, and 160. As shown in Figure 8, the surface generated through these linear combinations exhibited low loss and high accuracy, comparable to the perfor-

mance of the corner models. This success occurs because the corner models happen to be linearly connectable for every pair of them.

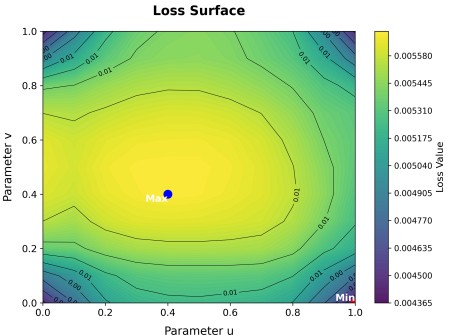
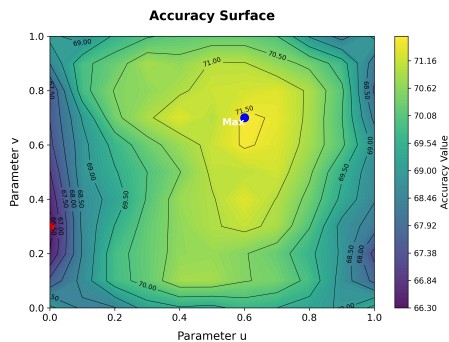

(a) Loss values on the Bézier surface. The surface linearly spans the four corner points, showing similarly low loss, matching the performance of the corner points. The corner points are ResNet18 models from epochs 220, 200, 180, and 160 during training on CIFAR-100.

(b) Accuracy values on the Bézier surface. This surface also spans the four corner points, showing high accuracy across epochs, matching the performance of the individual models.

Figure 8: Comparison of loss and accuracy surfaces on the Bézier surface spanned by four ResNet18 models, selected from epochs 220, 200, 180, and 160 during training on CIFAR-100. Both surfaces show consistency, with low loss and high accuracy distributed evenly across the surface, indicating that the corner models generalize well across the parameter space.

## A.3 EXISTENCE OF LINEAR SURFACE MODE CONNECTIVITY

While linear combinations of models often fail to produce a desirable surface, we have identified special cases where success is possible. Specifically, when $n = 4$ and each pair among these four models satisfies the linear mode connectivity property, a Bézier surface can be constructed in a linear manner where every model exhibits low loss and high accuracy. By selecting $\theta_{00}, \theta_{n0}, \theta_{0m}$, and $\theta_{nm}$ from a training trajectory, the pairs meet the linear mode connectivity condition. The interior control points $\theta_{ij}$ are obtained through linear interpolation of the four corner points:

$$\theta_{ij} = (1-t)(1-s)\theta_{00} + t(1-s)\theta_{n0} + (1-t)s\theta_{0m} + ts\theta_{nm},$$

where $t = \frac{i}{n}$ and $s = \frac{j}{m}$. The models on the surface are thus linear combinations of the initial models.

The Bézier surface $S(u, v)$ can be expressed as:

$$S(u,v) = C_{00}(u,v)P_{00} + C_{02}(u,v)P_{02} + C_{20}(u,v)P_{20} + C_{22}(u,v)P_{22},$$

where the coefficients $C_{ij}(u, v)$ are combinations of the basis polynomials and the relationships between control points.

**Computational Definition of the Basis Polynomials:** The Bernstein polynomials $B_{i,n}(u)$ are defined as:

$$B_{i,n}(u) = \binom{n}{i} u^i (1-u)^{n-i}, \quad i = 0, 1, \ldots, n.$$

Similarly, the polynomials $B_{j,m}(v)$ are defined as:

$$B_{j,m}(v) = \binom{m}{j} v^j (1-v)^{m-j}, \quad j = 0, 1, \ldots, m.$$

In our case, we have $n = m = 2$, so:

$$B_{0,2}(u) = (1-u)^2, \qquad B_{1,2}(u) = 2u(1-u), \qquad B_{2,2}(u) = u^2,$$
$$B_{0,2}(v) = (1-v)^2, \qquad B_{1,2}(v) = 2v(1-v), \qquad B_{2,2}(v) = v^2.$$

**Coefficients for Each Corner Control Point:** The coefficients $C_{ij}(u,v)$ are expressed in terms of the Bernstein polynomials:

$$C_{00}(u,v) = B_{0,2}(u)B_{0,2}(v) + \frac{1}{2}B_{0,2}(u)B_{1,2}(v) + \frac{1}{2}B_{1,2}(u)B_{0,2}(v) + \frac{1}{4}B_{1,2}(u)B_{1,2}(v),$$

$$C_{02}(u,v) = B_{0,2}(u)B_{2,2}(v) + \frac{1}{2}B_{0,2}(u)B_{1,2}(v) + \frac{1}{4}B_{1,2}(u)B_{1,2}(v) + \frac{1}{2}B_{1,2}(u)B_{2,2}(v),$$

$$C_{20}(u,v) = B_{2,2}(u)B_{0,2}(v) + \frac{1}{2}B_{1,2}(u)B_{0,2}(v) + \frac{1}{4}B_{1,2}(u)B_{1,2}(v) + \frac{1}{2}B_{2,2}(u)B_{1,2}(v),$$

$$C_{22}(u,v) = B_{2,2}(u)B_{2,2}(v) + \frac{1}{2}B_{1,2}(u)B_{2,2}(v) + \frac{1}{4}B_{1,2}(u)B_{1,2}(v) + \frac{1}{2}B_{2,2}(u)B_{1,2}(v).$$

One can verify that $C_{00}(u,v) + C_{02}(u,v) + C_{20}(u,v) + C_{22}(u,v) = 1$.

These coefficients sum to one, ensuring that $S(u,v)$ remains a convex combination of the corner points. We demonstrate this surface's effectiveness using VGG16 on the CIFAR-10 dataset, achieving low loss and high accuracy, comparable to that of the four corner points, as shown in Figure 6.

**Relationships Among the Coefficients** Although the coefficients $C_{00}(u,v), C_{02}(u,v), C_{20}(u,v)$, and $C_{22}(u,v)$ are functions of the parameters $u$ and $v$, they exhibit inherent relationships due to the properties of the Bernstein polynomials and the construction of the Bézier surface.

- **Sum-to-One Property:** For all $u, v \in [0, 1]$, the coefficients satisfy:

$$C_{00}(u,v) + C_{02}(u,v) + C_{20}(u,v) + C_{22}(u,v) = 1.$$

  This ensures that $S(u,v)$ is a convex combination of the corner control points, and the surface lies within their convex hull.

- **Symmetry Relations:** The coefficients exhibit symmetry properties due to the symmetrical nature of the Bernstein polynomials:

$$C_{00}(u,v) = C_{22}(1-u, 1-v),$$
$$C_{02}(u,v) = C_{20}(u, 1-v),$$
$$C_{20}(u,v) = C_{02}(1-u, v),$$
$$C_{22}(u,v) = C_{00}(1-u, 1-v).$$

  These relationships reflect the inherent symmetry in the Bézier surface construction.

- **Interdependence of Coefficients:** Any coefficient can be expressed in terms of the others using the sum-to-one property:

$$C_{ij}(u,v) = 1 - \sum_{\substack{(k,l) \\ (k,l) \neq (i,j)}} C_{kl}(u,v).$$

  This highlights the interconnectivity among the coefficients.

- **Proportional Relationships:** The ratios of certain coefficients are related through the parameters $u$ and $v$. For example:

$$\frac{C_{00}(u,v)}{C_{22}(u,v)} = \left(\frac{(1-u)(1-v)}{uv}\right)^2.$$

Similarly, the ratio between $C_{02}$ and $C_{20}$ is:

$$\frac{C_{02}(u,v)}{C_{20}(u,v)} = \left(\frac{(1-u)v}{u(1-v)}\right)^2.$$

These proportional relationships demonstrate how changes in $u$ and $v$ affect the relative influence of each corner control point on the surface.

## A.4 EXPERIMENTS ON SURFACES WITH MORE CONTROL POINTS

To further evaluate the scalability of our method, we conducted additional experiments on Bézier surfaces using 3×3, 4×4, and 5×5 control points on the CIFAR-10 dataset. These experiments were performed using a simple convolutional neural network (CNN) architecture, consisting of three convolutional layers with kernel size 3 and padding 1, followed by max-pooling layers for downsampling. The fully connected layers include a 256-dimensional hidden layer and a final output layer that matches the number of classes. Dropout is applied after the convolutional layers to mitigate overfitting.

These configurations allow us to test our method's ability to construct low-loss surfaces in increasingly complex parameter spaces. Table 2 summarizes the results of these experiments.

Table 2: Accuracy achieved with Bézier surfaces of different configurations on CIFAR-10.

| Control Points | Avg Acc of Corners (%) | Highest Acc (%) | Avg Acc for Surface (%) |
|---|---|---|---|
| 3×3 | 80.2 | 82.0 | 79.7 |
| 4×4 | 80.2 | 82.4 | 80.0 |
| 5×5 | 80.2 | 82.9 | 80.3 |

The results in Table 2 indicate that increasing the number of control points leads to better flexibility of the Bézier surface, resulting in better performance. Specifically, the highest accuracy achieved on the sampled points progressively increases as the number of control points grows from 3×3 to 5×5.

These findings demonstrate the robustness and scalability of our approach, showing its ability to capture low-training loss and high-test-accuracy regions in the parameter space, even under more complex configurations. This strengthens the potential for our method to extend to larger models and datasets in future work.

## A.5 MODEL ENSEMBLING COMPARISONS

To illustrate the advantages of our method for model ensembling, we conducted additional experiments comparing surface-based ensembling with traditional four-corner ensembling.

Table 3 summarizes the results of our experiments on CIFAR-10 with two architectures, VGG16 and ResNet18. The results highlight the advantage of surface-based ensembling in capturing additional diversity and achieving superior accuracy compared to four-corner ensembling.

Table 3: Comparison of four-corner ensembling and surface-based ensembling on CIFAR-10.

| Model | Four-Corner Ensemble (%) | Surface Ensemble (%) |
|---|---|---|
| VGG16 | 90.4 | 92.0 |
| ResNet18 | 90.1 | 92.7 |

The results demonstrate that surface-based ensembling consistently outperforms four-corner ensembling. By exploring the parameter space more comprehensively, Bézier surfaces capture diverse

model predictions, leading to improved accuracy during model ensembling. This advantage is particularly significant in scenarios where corner models reside in different basins, as traditional methods fail to navigate the non-linear loss landscape effectively.

These findings reinforce the applicability and robustness of our method for model ensembling, showcasing its potential in diverse configurations and datasets.

### A.6    LAYER-SPECIFIC OPTIMIZATION FOR EFFICIENCY

To improve computational efficiency, we conducted experiments where only a subset of the model's layers were optimized instead of the entire network. Specifically, we updated only the last convolutional layer and the fully connected layers in the Multi-Layer Perceptron (MLP) module. This approach significantly reduced computational overhead while maintaining competitive accuracy.

Due to time constraints, these experiments were validated on a simple convolutional neural network (CNN) architecture. The network consisted of three convolutional layers (kernel size 3, padding 1), max-pooling for downsampling, and a 256-dimensional fully connected hidden layer. The final layer matched the number of classes. The experiments were performed on a single NVIDIA 4090 GPU sampling 80 points per batch on the surface. The results are summarized in Table 4.

Table 4: Comparison of accuracy and efficiency for layer-specific optimization on CIFAR-10.

| Updated Layers | Avg Acc of Corners (%) | Highest Acc (%) | Avg Surface Acc (%) | Time (26 Epochs) |
|---|---|---|---|---|
| All layers | 80.2 | 82.0 | 79.7 | 47 min |
| Last conv + MLP | 80.2 | 80.3 | 70.3 | 15 min |

### RESULTS AND TRADE-OFFS

The results demonstrate that optimizing only the last convolutional and MLP layers yields significant efficiency gains, reducing runtime by over 68% with a single NVIDIA 4090 GPU, while incurring a moderate drop in surface accuracy. These findings highlight the potential of layer-specific optimization as a scalable approach for larger models and datasets.

## B    EFFICIENT EVALUATION OF LOSS AND ACCURACY SURFACES

### B.1    USING BATCH-LEVEL APPROXIMATION FOR TRAINING LOSS SURFACE

During training, 80 points are sampled per batch, covering most of the regions on the surface. Evaluating the losses for just the last few batches provides a reliable approximation of the full training loss on the surface. This observation suggests that a smaller subset of batches can effectively represent the overall surface, significantly reducing computation costs without sacrificing accuracy.

In our experiment on CIFAR-10 with the VGG16 architecture, we evaluated the training loss surface using a subset of data equivalent to four epochs. The results indicate strong consistency with evaluations conducted on the full training dataset. Specifically, when evaluated on the full training dataset, the valley of the loss surface was located at the $(u, v) = (0.9, 0.2)$. On the subset evaluation, this shifted slightly to $(0.9, 0.1)$. Both points, however, lie within the same low-loss region, underscoring the robustness of this approximation. Similarly, the peak of the loss surface consistently remained at $(u, v) = (0.4, 1.0)$ for both evaluations, as shown in Figure 9.

### B.2    TEST ACCURACY AND ITS APPROXIMATION

The grid search for the accuracy surface occurs during the inference phase and is thus far less computationally intensive than training. To further optimize efficiency, a subset of the test dataset can be used for evaluation. In our experiment, using CIFAR-10 with the VGG16 architecture, the accuracy surface evaluated on a subset of the test dataset showed a valley consistently located at $(u, v) = (0.4, 0)$, matching the results obtained from the full test dataset evaluation. The peak shifted slightly, from $(u, v) = (0.9, 0.2)$ to $(u, v) = (0.9, 0.1)$, but both points fall within the same high-accuracy region, demonstrating the surface's structural robustness.

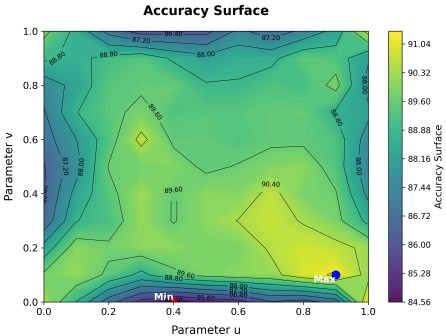
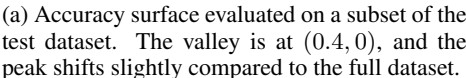

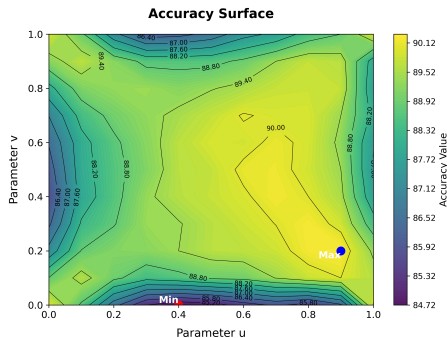

(a) Accuracy surface evaluated on a subset of the test dataset. The valley is at $(0.4, 0)$, and the peak shifts slightly compared to the full dataset.

(b) Accuracy surface evaluated on the full test dataset. The valley is at $(0.4, 0)$, and the valley remains same for both subset and full test datasets.

Figure 9: Comparison of accuracy surfaces evaluated on a subset (left) and the full test dataset (right) using CIFAR-10 with the VGG16 architecture. Both surfaces show consistent regions of high accuracy, demonstrating the reliability of subset evaluation.

