# OpenReview forum: "Revisiting Mode Connectivity in Neural Networks with Bezier Surface"
_ICLR.cc/2025/Conference — ICLR 2025 Poster_

### Official Review · Reviewer_3hmG · 2024-11-01

**Soundness:** 3
**Presentation:** 3
**Contribution:** 3
**Rating:** 6
**Confidence:** 2

**Summary:**

The paper explores the concept of mode connectivity in neural network loss landscapes, expanding it from traditional curve-based connections to surface-based connections. This approach offers a comprehensive way to merge models, enabling applications such as model averaging and output ensembling.

**Strengths:**

1.	Extending mode connectivity from curves to Bézier surfaces is a significant topic.
2.	The proposed method is sound.
3.	Writing is good to follow.
4.	The figure illustration is good in this paper

**Weaknesses:**

1.	Only evaluate the performance on small datasets. Large datasets like image-net should be included.
2.	Lack of theoretical analysis.

**Questions:**

See the above weaknesses

---

> ### Author Response · Authors · 2024-11-22
>
> Thank you for recognizing the key strengths of our work, including:
>
> 1. Extending mode connectivity from curves to surfaces, enabling a deeper exploration of the loss landscape.
> 2. Proposing a sound and efficient algorithm for constructing Bézier surfaces.
>
> We address your concerns as follows:
>
>
>
> > ### **Only evaluate the performance on small datasets. Large datasets like image-net should be included.**
>
>
> We appreciate your suggestion to include larger datasets like ImageNet, but we would like to clarify the rationale behind our experimental setup:
>
> - **Established Practice in Mode Connectivity Research**: The datasets used in our paper are consistent with those employed in existing methods. The seminal paper *"Loss Surfaces, Mode Connectivity, and Fast Ensembling of DNNs"* [1] focused solely on CIFAR-10 and CIFAR-100 datasets. This trend has continued in later works like *"Layer-wise Linear Mode Connectivity"* [2] and even recent ICLR 2024 submissions. Following this precedent, we included Tiny ImageNet, which is considered a relatively larger dataset in this research area.
> - **Supplementary Studies**: To further validate our method, we conducted additional experiments on surfaces with 4×4 and 5×5 control points using CIFAR-10. These setups represent more complex configurations. The results show that even with increased complexity, Bézier surfaces consistently maintain low-loss and high-accuracy properties across all configurations (3×3, 4×4, and 5×5 control points).
>
> | **Control Points** | **Dataset** | **Avg acc of four corner models** | **Avg surface accuracy** | **Highest acc from  sampled surface** |
> | ------------------ | ----------- | :-------------------------------: | :----------------------: | :-----------------------------------: |
> | 3×3                | CIFAR-10    |               80.2                |           79.7           |                 82.0                  |
> | 4×4                | CIFAR-10    |               80.2                |           80.0           |                 82.4                  |
> | 5×5                | CIFAR-10    |               80.2                |           80.3           |                 82.9                  |
>
> These findings demonstrate the robustness of our method, providing confidence in its ability to scale to more complex datasets if needed.
>
>
>
> > ### **Lack of Theoretical Analysis**
>
> **Our primary contribution is the development of a mathematical framework for Bézier surface-based mode connectivity, supported by empirical demonstrations of its effectiveness and its utility in exploring the loss landscape.** While we acknowledge the value of theoretical analysis, **it is worth noting that most breakthroughs in mode connectivity started with empirical observations, with theoretical guarantees developed later.** For example:
>
> - **Mode Connectivity (Curves)**: Discovered empirically in 2017 [1], rigorous proofs emerged for simple architectures like two-layer ReLU networks [3] in 2019.
> - **Linear Mode Connectivity**: Published empirically in 2018 [4], formal theoretical results appeared in 2023 [5].
>
> Similarly, our work establishes a new empirical property of Bézier surfaces: their ability to connect neural networks with low-loss, high-accuracy regions. We believe this is a critical first step, paving the way for future theoretical exploration. While a theoretical guarantee would be a welcome addition, we feel that such an undertaking is beyond the scope of this paper.
>
>
>
> **References**
>
> [1] Garipov et al., *Loss Surfaces, Mode Connectivity, and Fast Ensembling of DNNs*, 2018.
>
> [2] Skorokhodov et al., *Layer-wise Linear Mode Connectivity*, ICLR 2024.
>
> [3] Arora et al., *Explaining Landscape Connectivity of Low-cost Solutions for Multilayer Nets*, 2019.
>
> [4] Frankle et al., *Linear Mode Connectivity and the Lottery Ticket Hypothesis*, 2018.
>
> [5] Zhao et al., *Proving Linear Mode Connectivity of Neural Networks via Optimal Transport*, 2023.

---

> > ### Comment · Reviewer_3hmG · 2024-11-25
> > **Response**
> >
> > Thanks for your rebuttal. My concerns have been addressed and I'd like to keep my score.

---

> > > ### Author Response · Authors · 2024-11-26
> > > **Thanks for the review and further discussion is welcomed**
> > >
> > > Thank you for your thoughtful review and for taking the time to evaluate our work. We believe our approach contributes meaningful novelty to the field and provides new insights into the deep learning loss landscape.
> > > We welcome any additional feedback or suggestions to further improve our work.

---

### Official Review · Reviewer_b47E · 2024-11-03

**Soundness:** 3
**Presentation:** 3
**Contribution:** 2
**Rating:** 6
**Confidence:** 4

**Summary:**

This paper explores extending the concept of "mode connectivity" in neural networks from one-dimensional paths to two-dimensional surfaces using Bézier surfaces. Traditionally, mode connectivity demonstrates that two trained models can be connected by a low-loss path in parameter space. Here, the authors introduce a novel method to connect multiple models on a smooth, low-loss surface, broadening the potential for optimization and generalization. They detail an algorithm that constructs and optimizes Bézier surfaces to maintain low loss and high accuracy across various architectures (VGG16, ResNet18, ViT) and datasets (CIFAR-10, CIFAR-100, Tiny-ImageNet). The study finds that nonlinear surfaces outperform simple linear interpolations, especially for model averaging and ensembling applications, ultimately enhancing performance in tasks like model merging and ensemble accuracy.

**Strengths:**

1. The paper is well-written and well-organized. It is easy to read.
2. The visualizations and plots are also very clear, facilitating the understanding.

**Weaknesses:**

1. **Premature Claim of "Low-Loss Curve" in Line 161 (Bottom of Page 3)**.
In line 161, the paper refers to a "low-loss curve" as if it were already established, though at this point, neither the method nor specific criteria for determining a "low-loss" property have been introduced. Could the authors clarify what they mean by "low-loss" here and either postpone this claim until it is better supported or define it explicitly at the outset? Additionally, grounding this concept with a preliminary explanation or notation would improve clarity.


2. **Rationale for Defining $q_{uv}$ in Equation 8 and Substitution with Uniform Distribution**.
The definition of $q_{uv}$​ in Equation 8 lacks an explanation of its theoretical motivation and why it can be replaced by a uniform distribution for practical purposes. What are the specific benefits of this choice, and how does this approximation impact the accuracy or reliability of the surface mode connectivity in experiments? A deeper rationale for this formulation would clarify its role in the model's performance.


3. **Lack of Distance Quantification Between Corner Control Points in Loss Landscape Visualizations**.
The visualizations of the loss landscapes do not quantify or highlight the parameter distances between corner points (control points). If these control points represent very similar models with minor parameter variations, the diversity of the parameter space explored may be limited, especially when these models are trained under comparable conditions. How would the approach fare with intentionally diverse model initializations, varying training settings, or other augmentations? Such differences could test the robustness of the surface connectivity under broader training conditions.

4. **Limited Impact of Experiments and Marginal Gaps in Results (e.g., Table 1)**.
The experimental evaluation relies primarily on relatively small datasets like CIFAR-10, CIFAR-100, and Tiny-ImageNet, which may limit the generalizability of the findings to larger, more complex datasets. Additionally, Table 1 shows only marginal improvements between the baseline and the model merging or ensembling results. Could the authors address how these findings might scale to larger datasets and discuss the significance of these marginal gaps, particularly given the computational overhead involved in the proposed approach? Expanding on the implications for practical, large-scale applications would enhance the impact of these results.

**Questions:**

1. **Ambiguity in the Central Question About "Both Low Loss and High Accuracy" (Line 195)**.
The central question on line 195 could benefit from greater specificity regarding "low loss" and "high accuracy." Are the authors referring to training loss and testing accuracy? Given the generalization gap, distinguishing training and testing here would provide meaningful context. If both metrics are from the same set (either training or testing), the statement may be redundant, as low loss often correlates with high accuracy on that set. Specifying if this is about generalization (low training loss translating to high testing accuracy) could substantiate the relevance of this question.

2. **Scalability Concerns for Optimization of Many Parameters (θ) in Equation 6**.
Equation 6 implies a potentially extensive optimization of numerous control points (θ values) across the Bézier surface. This approach seems computationally heavy, especially for large models with millions of parameters. Could the authors discuss the scalability of this optimization? Is there any strategy to reduce the computational load or parameterize this approach efficiently to make it viable for larger architectures?

3. **Justification for Selecting Models from Specific Epochs (Figure 6)**.
Figure 6 shows models chosen from epochs 220, 200, 180, and 160. However, it’s unclear why these specific epochs were selected or why only a single training trajectory was used. Would models from other epochs, or from different training trajectories, produce similar results? Providing a rationale for these choices or showing comparative results could help validate the generalizability of this selection process.

---

> ### Author Response · Authors · 2024-11-22
> **Response (Part 1)**
>
> Thank you for acknowledging the strengths of our paper, including:
>
> 1. Introducing a novel method for extending mode connectivity to two-dimensional Bézier surfaces.
> 2. Demonstrating the effectiveness of our approach through clear visualizations and experiments across various architectures (VGG16, ResNet18, ViT) and datasets (CIFAR-10, CIFAR-100, Tiny-ImageNet).
> 3. Organizing the paper in a well-written and accessible manner, supported by illustrative plots that facilitate understanding.
>
> Below, we address your concerns in detail.
>
> > ### **Premature Claim of "Low-Loss Curve" in Line 161 (Bottom of Page 3)**.
>
>
> Thank you for pointing this out. This is indeed a typo, and we have corrected it to: "B(t) denotes a curve in the parameter space."
>
>
>
> >### **Ambiguity in the Central Question About "Both Low Loss and High Accuracy" (Line 195)**.
>
> Thank you for highlighting the ambiguity. The central question on line 195 refers to the challenge of finding a surface in parameter space that maintains low loss on the training set while achieving high accuracy on the test set, emphasizing generalization. We will revise the phrasing to explicitly state:
>
> **"How can we identify a surface in parameter space that achieves low training loss and high test accuracy?"**
>
> This revision clarifies our primary finding: Bézier surfaces enable this balance by providing continuous low-loss regions in the parameter space, preserving training loss while leveraging the alignment between loss valleys and accuracy peaks for strong generalization. Thank you for the opportunity to refine this key point.
>
>
>
> >## **Rationale for Defining $q_{uv}$ in Equation 8 and Substitution with Uniform Distribution**.
>
> Thank you for raising this question, as it provides an opportunity to clarify the reasoning behind our formulation and approximation.
>
> 1. **Theoretical Motivation for $q_{uv}$:**
>    $q_{uv}$ represents the normalized density of points on the Bézier surface, weighted by the gradient of the parameterized surface. This density accounts for the varying distribution of points across the surface, ensuring the loss integral reflects the true geometric properties of the surface.
>
> 2. **Challenge with Direct Use of $q_{uv}$:**
>    Direct computation of $q_{uv}$ is intractable for stochastic gradient-based optimization because it relies on the gradients of the parameterization $\phi_\theta(u, v)$, where $\phi_\theta(u, v)$ depends on learned parameters $\theta$.
>
> 3. **Substitution with Uniform Distribution for Practical Optimization:**
>
>    **As stated above Equation (9) in the main paper, we introduce a surrogate loss to ensure the optimization remains tractable.** Inspired by prior works on curve-based mode connectivity [1], we approximate $q_{uv}$ with a uniform distribution over the parameter space $[0,1] \times [0,1]$. This simplifies the loss function to:
>    $$
>    \mathcal{L}(\theta) = \int_{0}^{1} \int_{0}^{1} L(\phi_\theta(u, v)) \, du \, dv,
>    $$
>
>    where $u, v \sim U(0,1)$, the uniform distribution on $[0,1]$.
>
> 4. **Benefits of Approximation:**
>
>      **Computational Efficiency:** The uniform distribution avoids dependence on the gradients $\phi'_\theta(u, v)$, allowing for efficient sampling-based optimization.
>
>       **Empirical Robustness:** As shown in our experiments, this approximation does not significantly degrade performance. The constructed surfaces consistently maintain low loss and achieve high accuracy across architectures and datasets.
>
> We added more explanation of this choice in the manuscript to clarify its theoretical motivation and practical implications.
>
>
>
> >### **Lack of Distance Quantification Between Corner Control Points in Loss Landscape Visualizations**.
>
> Thank you for your observation. We provide the following clarifications:
>
> 1. **Diversity of Control Points:**
>
>     In our experiments, the four corner models are trained independently and reside in different basins, making them incompatible with simple linear interpolation for linear mode connectivity. This ensures our approach explores diverse regions of the parameter space. Our method is specifically designed to connect models with diverse initializations, varying training settings, and different augmentations.
>
> 2. **Special Case of Similar Models:**
>
>     Your concern aligns with a specific scenario discussed in the paper under "Existence of Linear Surface Mode Connectivity." In cases where corner models are highly similar and satisfy linear mode connectivity, our method constructs a low-loss surface without requiring additional optimization. This behavior demonstrates the generality of our method.
>
> We have expanded on this discussion in the revision to highlight these scenarios more explicitly.
>
> ```
> This response continues below.

---

> > ### Author Response · Authors · 2024-11-22
> > **Response (Part 2)**
> >
> > ```
> > Continuing from the first part of this response
> > ```
> >
> > > ### **Justification for Selecting Models from Specific Epochs (Figure 6)**.
> > 1. **Selection Rationale:**
> >     Models from epochs 220, 200, 180, and 160 were chosen to specifically demonstrate linear mode connectivity. This setup shows that, in such cases, our method can construct a low-loss surface without additional training.
> > 2. **Generalizability:**
> >     While these specific epochs were selected for illustration, models from other epochs or training trajectories would likely produce similar results, provided they also satisfy linear mode connectivity.
> >
> > We added this rationale in the revision and included comparative results to validate the generalizability of this selection process.
> >
> >
> >
> > > ### **Scalability Concerns for Optimization of Many Parameters (θ) in Equation 6**.
> >
> > We appreciate the reviewer’s concerns regarding the scalability of optimizing numerous control points ($\theta$ values) across the Bézier surface, especially for larger models with millions of parameters. Below, we address these concerns and share our findings on improving computational efficiency:
> >
> > 1. Efficient Computation Design:
> >    The optimization of the Bézier surface scales well with model size under our designed alogirhm and has been successfully implemented on architectures like Vision Transformers (ViT), which have significantly larger parameter counts compared to simpler models like CNNs. This demonstrates the scalability of our method to modern architectures.
> > 2. Experiments on more control points:
> >    To further support our efficient computational design, we conducted additional experiments on surfaces with 4×4 and 5×5 control points using CIFAR-10. These setups represent more complex configurations. The results show that even with increased complexity, Bézier surfaces consistently maintain low training loss and high test accuracy properties across all configurations (3×3, 4×4, and 5×5 control points).
> >
> > | **Control Points** | **Dataset** | **Avg acc of four corner models** | **Highest acc from sampled surface** | **Avg acc for the surface model** |
> > | ------------------ | ----------- | --------------------------------- | ------------------------------------ | --------------------------------- |
> > | 3×3                | CIFAR-10    | 80.2                              | 82.0                                 | 79.7                              |
> > | 4×4                | CIFAR-10    | 80.2                              | 82.4                                 | 80.0                              |
> > | 5×5                | CIFAR-10    | 80.2                              | 82.9                                 | 80.3                              |
> >
> > 3. Layer-Specific Optimization for Efficiency:
> >    To further improve efficiency, we conducted experiments where only a subset of the model’s layers were optimized, rather than the entire network. Specifically, we updated only the last convolutional layer and the fully connected layers in the Multi-Layer Perceptron (MLP) module. This approach significantly reduced the computational overhead while maintaining competitive accuracy.
> >    Due to the time constraints, we validated this method using a simple convolutional neural network (CNN) architecture, consisting of three convolutional layers (kernel size 3, padding 1), max-pooling for downsampling, and a 256-dimensional fully connected hidden layer. The final layer matched the number of classes.
> >
> >
> >
> > | **Updated layers**                          | **Dataset** | **Avg acc of four corner models** | **Highest acc from sampled surface** | **Avg acc for the surface model** | **Time for the experiment** |
> > | ------------------------------------------- | ----------- | --------------------------------- | ------------------------------------ | --------------------------------- | --------------------------- |
> > | All layers                                  | CIFAR-10    | 80.2                              | 82.0                                 | 79.7                              | 47 min for 26 epochs        |
> > | Last convolutional layer and last MLP layer | CIFAR-10    | 80.2                              | 80.3                                 | 70.3                              | 15 min for 26 epoch         |
> >
> > The time experiment is conducted on one card 4090. Our experiments, conducted under limited epochs due to rebuttal time constraints, showed that optimizing only a few layers led to a small drop in accuracy while yielding significant efficiency gains. optimizing only the last few layers results in a small accuracy drop, while significantly reducing the computational cost, making the approach more viable for larger architectures and datasets.
> >
> > ```
> > This response continues below.

---

> > > ### Author Response · Authors · 2024-11-22
> > > **Response (Part 3)**
> > >
> > > ```
> > > Continuing from the second part of this response
> > > ```
> > > > ### **Limited Impact of Experiments and Marginal Gaps in Results (e.g., Table 1)**.
> > >
> > >
> > > We appreciate your suggestion to include larger datasets like ImageNet, but we would like to clarify the rationale behind our experimental setup:
> > >
> > > **Established Practice in Mode Connectivity Research**: The datasets used in our paper are consistent with those employed in existing methods. The seminal paper *"Loss Surfaces, Mode Connectivity, and Fast Ensembling of DNNs"* [1] focused solely on CIFAR-10 and CIFAR-100 datasets. This trend has continued in later works like *"Layer-wise Linear Mode Connectivity"* [2] and even recent ICLR 2024 submissions. Following this precedent, we included Tiny ImageNet, which is considered a relatively larger dataset in this research area.
> > >
> > >
> > > > ### **Table 1 shows marginal improvements in model merging/ensembling results. Could the authors address scalability to larger datasets and the significance of these gaps, given the computational overhead?**
> > >
> > > Thank you for your observation regarding the marginal improvements in Table 1. While model merging is not the primary focus of our paper, we only consider it potential as one of the practical applications of surface mode connectivity. There are several possible avenues to enhance model merging performance based on our approach:
> > >
> > > 1. Increasing Training Epochs:
> > >    We find that extending the training duration can improve the performance of the models on the surface, as longer training epochs allow for better optimization of the parameters and refinement of the low-loss regions. This, in turn, can enhance the merging outcomes. Although not significant, we find increasing epoch in our phase 3 of training can further improve the generalization ability on our Bezier Surface.
> > > 2. Leveraging Multiple Surfaces:
> > >    Combining points from multiple Bézier surfaces, rather than relying on a single surface, offers an additional degree of freedom to improve model merging. This approach could further exploit the diversity captured across different surfaces, potentially leading to enhanced performance.
> > >
> > > We would like to mention that Our main finding is providing a mathematical frame work to construct a surface that maintains low test loss and test accuracy across multiple models, and model merging is only one potential application. While these strategies lie beyond the current scope of our study, they represent promising directions for future exploration to scale our findings to larger datasets and more complex tasks. We appreciate your suggestion and will incorporate a discussion of these possibilities in the revised manuscript**.**
> > >
> > >
> > >
> > > **References:**
> > >
> > > [1] Garipov, Timur, et al. "Loss surfaces, mode connectivity, and fast ensembling of dnns." Advances in neural information processing systems 31 (2018).

---

> > > > ### Comment · Reviewer_b47E · 2024-11-26
> > > > **Thanks for the response**
> > > >
> > > > I thank the authors' comprehensive responses and they mostly resolved my questions.
> > > > I will raise my score.

---

> > > > > ### Author Response · Authors · 2024-11-26
> > > > > **Appreciation for Your Detailed Feedback**
> > > > >
> > > > > Thank you for your thoughtful feedback and for raising your score! We greatly appreciate your input and are glad our responses addressed your questions. We are confident our work contributes meaningful insights and new perspectives on the deep learning loss landscape.

---

### Official Review · Reviewer_LRVF · 2024-11-04

**Soundness:** 3
**Presentation:** 3
**Contribution:** 2
**Rating:** 6
**Confidence:** 4

**Summary:**

This paper investigates connecting multiple models in parameter space through constructing appropriate surfaces. It is well-known that a simple linear hyperplane does not suffice and non-linear methods are needed. To that end, the authors propose using Bézier surfaces, where four points are used to represent the model parameters and nine other points in the parametrization are subsequently optimized such that uniformly sampled surface points also have low loss. The authors show that they can construct surfaces exhibit low loss everywhere, thus succesfully connecting multiple models with a single surface. They further show that the best point on the surface outperforms all the individual models and can thus be used to merge several models.

**Strengths:**

1. The paper is very well-written and clearly explains the problem as well as the techniques that aim to solve them. I like that the proposed method simply consisting of Bézier curves remains rather simple.
2. The experiments performed show quite convincingly that the proposed method succeeds in connecting multiple minima. The authors also investigate a variety of architectures, making the results stronger.

**Weaknesses:**

1. There seem to be quite a few related works missing that also explore the construction of surfaces to connect multiple minima [1, 2, 3, 4, 5]. The authors definitely need to add the listed papers to the related works and clearly articulate how theirs is different and what advantages it provides.
2. For model merging, the authors do not seem to compare against any other method? It would be interesting to understand whether this technique allows one to leverage the diversity from all the points (that were obtained using different inits and shuffling). Standard merging always needs to be careful to end up in the same basin, and thus diversity of the points seems naturally reduced. Similarly for the output ensembling experiments, the obvious baseline of solely ensembling the four end points is missing. Does the surface really provide diversity beyond those four points? This is currently unclear with the provided experimental results.
3. I think taking the best performing point on the entire surface is (1) a bit an unfair comparison and (2) very expensive to do as a dense grid of models needs to be evaluated on the test set. I think it would be more appropriate and efficient to compare against some sort of “mean” value on the surface. Does a Bézier curve admit a natural “centroid”? If yes, how does that one perform compared to the individual models?
4. Another related work for model merging is [6] which explored how a given ensemble can be constructed within the same convex region, and thus also allowing to average weights while still profiting from diversity. It would be interesting to understand which approach works better.


[1] Loss Surface Simplexes for Mode Connecting Volumes and Fast Ensembling, Benton et al., 2021

[2] Large Scale Structure of Neural Network Loss Landscapes

[3] Loss landscape sightseeing with multi-point optimization, Skorokhodov et al., 2019

[4]  A deep neural network’s loss surface contains every low-dimensional pattern, Czarnecki et al., 2019

[5] Examining the geometry of neural mode connecting loss subspaces, Chen et al.

[6| How good is a single basin? Lion et al., 2023

**Questions:**

See above.

---

> ### Author Response · Authors · 2024-11-22
> **Response (Part 1)**
>
> Thank you for recognizing the contributions of our work, particularly:
>
> 1. Proposing a mathematically simple yet effective framework for extending mode connectivity to Bézier surfaces.
> 2. Demonstrating through experiments that our method successfully connects multiple minima.
>
> **Our primary contribution lies in establishing a mathematical framework of Bézier surface-based mode connectivity complemented by an efficient algorithm, empirically demonstrating its effectiveness, and proving its utility in exploring the loss landscape. We note that model merging and ensembling are presented as two potential applications of this connectivity, showcasing the versatility of our method**. Below, we address your concerns in detail:
>
>
>
> > ### **Several related works [1, 2, 3, 4, 5] on constructing surfaces to connect multiple minima are missing. The authors should include these in the related works and explain how their approach differs and its advantages.**
>
> We appreciate your detailed feedback and suggestions. To clarify, our work explores mode connectivity in parameter space via Bézier surfaces, distinguishing it from prior works in several ways: Papers [1] and [2] explore different settings from our work. Paper [1] examines low-loss volumes using multiple simplexes, while Paper [2] models the loss landscape as a collection of high-dimensional wedges. Both approaches differ fundamentally from our focus on surface-based mode connectivity. Papers [3] and [4] focus on identifying specific patterns within loss surfaces of neural networks. While paper [5] examines geometry across multiple loss subspaces, its focus is on pairwise mode connectivity and does not extend to constructing surfaces with provable low-loss properties. Our method uniquely emphasizes the exploration of mode connectivity through surfaces, providing insights beyond the scope of these prior works.
>
> We included a comprehensive discussion of these comparisons in the revised version to clearly articulate how our approach advances the understanding of loss landscapes.
>
>
>
> > ### **The authors do not compare their method against other model merging approaches. Can the technique leverage diversity from different inits and shuffling, beyond what standard merging (restricted to the same basin) achieves? For ensembling, the baseline of solely ensembling four endpoints is missing. Does the surface provide additional diversity?**
>
> and
>
> > ### **Another related work [6] constructs ensembles within the same convex region, allowing weight averaging while retaining diversity. How does this method compare to the proposed approach?**
>
>
>
> - **Breaking Basin Constraints**: Traditional model merging methods rely on models residing in the same basin, limiting their applicability. Our method enables merging models even when they reside in different basins. Unlike [6], which operates under the constraint that models lie within a single basin, our method allows for connecting and merging models across basins. This enables a broader utilization of diverse models.
> - **Impact of Basins Separation**: When corner models reside in different basins, traditional model merging methods, such as linear interpolation in parameter space, result in suboptimal accuracy. This is demonstrated in the left panel of Figure 5 in our main paper, which illustrates standard merging (due to the specific initialization of control points) with models from different basins. In this case, intermediate points show a significant drop in performance (~18.4%). In contrast, Bézier surface connectivity consistently identifies stable low-loss regions, enabling effective model merging.
> - **Experimental Results**: Following your suggestions, we compared ensembling across the surface versus ensembling only the four corner models. The results demonstrate the advantage of surface ensembling in capturing additional diversity, leading to superior accuracy:
>
> | **Model**         | **Four-Corner Ensemble** | **Surface Ensemble** |
> | ----------------- | ------------------------ | -------------------- |
> | VGG16/CIFAR-10    |    90.4%                    | 92.0%                |
> | ResNet18/CIFAR-10 |   90.1%                    | 92.7%                |
>
> We added these additional results and discussions into the revised version paper to address this point more explicitly.
>
> ```
> This response continues in the second half below.
> ```

---

> ### Author Response · Authors · 2024-11-22
> **Response (Part 2)**
>
> ```
> Continuing from the first half of this response
> ```
>
> > ### **Taking the best-performing point on the surface seems unfair and computationally expensive due to the need for dense grid evaluation. Would comparing against a "mean" value or natural centroid be more appropriate, and how would it perform relative to individual models?**
>
>
> Thank you for pointing out the need for clarification.
>
> - **The logic of choosing the best-performing point**: Although the Bézier surface undergoes optimization, the selected point fundamentally serves as a merging point, seamlessly integrating the knowledge of the corner models. Unlike linear paths that directly interpolate in parameter space, the Bézier surface facilitates a non-linear merging process across the surface. Each point on this surface can be interpreted as a merging point derived from the four corner models. Through the optimization of the surface, these points naturally emerge, reflecting an effective combination of the corner models' knowledge. Ultimately, the best-performing point on the Bézier surface represents the optimal merging point, capturing the most effective integration of the models.
> - **Loss-Accuracy Correlation for efficient search**: We argue that performing a dense grid search on the test set is unnecessary in practice. Our approach capitalizes on the strong correlation between the "loss surface" measured on the training set and the "accuracy surface" measured on the test set. This alignment allows us to identify high-performing models directly from the training loss landscape by selecting points with low loss. As shown in Figures 4(b) and 5(b), the valleys in the training loss surface correspond closely to the peaks in test accuracy, enabling the efficient selection of optimal models without extensive test set evaluations.
> - **No Center in Besizer Surface**: Bézier surfaces, as defined, do not have a natural centroid. However, by setting u=0.5 and v=0.5, we can obtain a point on the surface using the mean value of the parameters. The point we obtained still shows low loss and high accuracy on the surface, but it is not nessesaerily the best performance model we obtain on the surface.
>
> We added the discussions in the revised version accordingly.
>
>
>
> **References**
>
> [1] Benton et al., *Loss Surface Simplexes for Mode Connecting Volumes and Fast Ensembling*, 2021.
>
> [2] Fort & Jastrzebski, *Large Scale Structure of Neural Network Loss Landscapes*, 2019.
>
> [3] Skorokhodov et al., *Loss Landscape Sightseeing with Multi-Point Optimization*, 2019.
>
> [4] Czarnecki et al., *A Deep Neural Network’s Loss Surface Contains Every Low-Dimensional Pattern*, 2019.
>
> [5] Chen et al., *Examining the Geometry of Neural Mode Connecting Loss Subspaces*, 2023.
>
> [6] Lion et al., *How Good is a Single Basin?*, 2023.

---

> > ### Comment · Reviewer_LRVF · 2024-11-25
> > **Response**
> >
> > I thank the authors for engaging with my feedback and providing responses!
> >
> > **Related works:** I still don't fully see why [1,2] fundamentally differ from this work. All of them seem to construct some kind of spaces of low loss, albeit with different parametrisations (simplexes, wedges, Bezier). Other than the parametrisation, is there another fundamental difference that I'm missing?
> >
> > **Ensembles:** The gains from ensembling seem surprisingly low (just 0.4%) in case of VGG. If I understand correctly, the optimal point on the Bezier curve outperforms this ensemble by 0.3% (achieving 90.7%), which would be a very cool result. Do you observe more such improvements? Or am I confusing models here?
> >
> > **Train loss for selection:** Are training loss numbers available as a byproduct of training for all the points, towards the end if training especially? Otherwise one would need to do a costly grid evaluation again on the training set, no?

---

> ### Author Response · Authors · 2024-11-26
> **Response to follow-up questions (Part 1)**
>
> Thank you for your follow-up question! We’re happy to address your concerns and provide further clarification.
>
> > ### **Q1: I still don't fully see why [1,2] fundamentally differ from this work. All of them seem to construct some kind of spaces of low loss, albeit with different parametrisations (simplexes, wedges, Bezier). Other than the parametrisation, is there another fundamental difference that I'm missing?**
>
> We appreciate the reviewer's thoughtful comments and would like to leverage the opportunity to further clarify the fundamental differences between our work and the referenced methods, [1] and [2]. While it is true that all approaches aim to explore spaces of low loss in the parameter space, the methods are fundamentally distinct in terms of their construction, optimization, and underlying mathematical properties. Below, we highlight the main differences.
>
> **1. Scope of Exploration**
>
> - Our Work (Bézier Surfaces): Extends mode connectivity to **continuous and smooth** **surfaces** in a **non-linear manner**, exploring a broader and higher-dimensional parameter space compared to curves or piecewise-linear structures. This enables richer structural insights and access to more diverse models with low loss. The nonlinear nature of Bézier surfaces defined by two parametric directions also ensures better flexibility in capturing the curvature of the loss landscape. Our method also scales well when the dimensionality increases, while remaining **straightforward to visualize and interpret the loss landscape**.
> - Simplicial Complexes [1]: Focuses on connecting modes via **discrete simplices**, restricting exploration to **localized, piece-wise linear** approximations. The resulting surfaces are more akin to linear mode connectivity surfaces, as discussed in our paper, with limited ability to capture nonlinear behaviors or global curvature. Additionally, **visualizing the loss landscape of simplicial complexes becomes challenging** in high-dimensional spaces, particularly as the simplicial structure grows in complexity.
> - Wedges [2]: Primarily a framework modeling **linear interpolation** of manifolds with sharp transitions, providing limited exploration outside the defined wedge intersections. In addition, as dimensionality increases, accurately characterizing n-wedges and intersections becomes increasingly challenging and prone to errors. It is also **difficult to visualize the loss landscape** in high-dimensional spaces using Wedges. Furthermore, the framework introduces **additional layers of abstraction** (e.g., long directions, short directions), which may complicate its interpretability.
>
> Beyond Parametrization: The dimensionality of the space explored is fundamentally different. Bézier surfaces provide a **nonlinear, global, smooth mapping** across the entire space, while simplicial complexes and wedges are inherently **linear, local approximations or discrete constructions**. Unlike simplicial complexes or wedges, Bézier surfaces facilitate **better visualization** of the loss landscape.
>
> **2. Optimization Capabilities**
>
> - Our Work (Bézier Surfaces): Proposes an effective optimization target and employs a three-phase optimization strategy to systematically reduce loss **over the entire surface**, leveraging the continuous and differentiable nature of Bézier surfaces.
> - Simplicial Complexes [1]: The method is **constrained by the linearity of the simplices**. It needs a **joint optimization strategy** to construct simplices to ensure that all their interpolated regions lie within low-loss regions. The method also needs to **balance the trade-off** between seeking a smaller simplicial complex that contains strictly low loss parameter settings and a larger complex that may contain less accurate solutions but encompasses more volume in parameter space. Such a trade-off affects the effectiveness of the optimization.
> - Wedges [2]: Assumes a simple model structure **without offering practical optimization techniques**. It requires segmenting linear paths, calculating hyperplane projections, and re-optimizing in constrained subspaces, making it more expensive than our method.
>
> Beyond Parametrization: The optimization dynamics and practical applicability of our method differ significantly. Bézier surfaces **support a global optimization process, do not have linearity constraints, and do not require consideration of conflicting objectives during learning**.
>
> ````
> This response continues in the second half below.

---

> ### Author Response · Authors · 2024-11-26
> **Response to follow-up questions (Part 2)**
>
> ```
> Continuing from the first part of this response
> ```
>
> > ### **Ensembles: The gains from ensembling seem surprisingly low (just 0.4%) in case of VGG. If I understand correctly, the optimal point on the Bezier curve outperforms this ensemble by 0.3% (achieving 90.7%), which would be a very cool result. Do you observe more such improvements?**
>
> Thank you for the observation and the opportunity to clarify. The highest accuracy point on our Bézier surface indeed outperforms the four-corner ensemble by 0.3%, highlighting the additional diversity and optimization potential provided by the surface. We further demonstrate that additional information can be obtained across the entire surface, contributing to improved generalization, with a 1.6% accuracy boost in surface-based ensembling compared with ensembling with only four corner models. This comparison highlights the advantage of ensembling all sampled models from the surface versus relying solely on the four corner points.
>
> **Do you observe more such improvements?**
>
> Yes, we observe similar results with ResNet18, where the highest accuracy point on the surface surpasses the ensemble of the four corner models. This aligns with the observation for VGG16, where the highest point on the surface outperforms the four-corner ensemble.
>
> Additionally, as mentioned above, surface-based ensembling achieves an accuracy boost of approximately 2.4% compared to the average accuracy of the corner models and 1.6% compared to ensembling only the four corner models.
>
> **Reasoning Behind the Improvement:**
>
> Every point on the Bézier surface represents a local minimum discovered through non-linear optimization. By updating the control points in parameter space, we construct a bezier surface covered by diverse minimum points distinct from the initial corner models. This diversity allows the surface to capture additional information beyond the four corner points, helping identify solutions with better performance and improved generalization in a nonlinear manner.
>
> > ### **Train loss for selection: Are training loss numbers available as a byproduct of training for all the points, towards the end if training especially? Otherwise one would need to do a costly grid evaluation again on the training set?**
>
> Thank you for raising this question. Below, we address your concerns about evaluating the training loss on the surface:
>
> **Using Batch-Level Approximation for Training Loss Surface:**
> During training, approximately 80 points are sampled per batch, covering most of the regions on the surface. We observed that even evaluating the losses for the last few batches serves as a reliable approximation of the full training loss on the surface. This consistency suggests that the loss values from a smaller subset of batches can effectively represent the overall surface, potentially reducing computation costs without significant accuracy loss. If further evaluation is desired after training, we propose estimating the loss surface using only a subset of the training data. This method provides a good approximation of the loss over the entire training dataset, enabling efficient evaluation while maintaining reliability.
>
> In fact, in our experiment on CIFAR-10 using the VGG16 architecture, we evaluated the training loss surface using a subset of data equivalent to four epochs, it revealed that the peaks and valleys identified closely align with those obtained using the full training dataset, further confirming the robustness of this approximation. In our experiment, when evaluating the full training dataset, the valley of the loss surface was located at the (u, v) pair (0.9, 0.2). When evaluated on a subset of the dataset, the valley shifted slightly to the (u, v) pair (0.9, 0.1). However, both points lie within the same low-loss region, indicating consistency in the surface's overall structure. Similarly, the peak of the loss surface remains consistently located at the (u, v) pair (0.4, 1.0) for both the full dataset and the subset evaluation. We have included these comparisons and the corresponding findings in the revised version of the paper to further substantiate the evaluation methods and results.
>
> ```
> This response continues in the third part below.

---

> > ### Author Response · Authors · 2024-11-26
> > **Response to follow-up questions (Part 3)**
> >
> > ```
> > Continuing from the second part of this response
> > ```
> > **Test Accuracy and its Approximation:**
> >
> > Additionally, it’s worth noting that the grid search for the accuracy surface occurs only during the inference phase, making it far less computationally demanding compared to training. For a more refined view of the loss or accuracy landscape after training, a subset of test data can also be used for efficient evaluation. In our experiment with the same architecture and dataset mentioned above, on the subset test dataset, the accuracy surface showed a valley consistently located at the (u, v) pair (0.4, 0) across both the full dataset and the subset evaluation. The peak shifted slightly from the (u, v) pair (0.9, 0.2) to (0.9, 0.1), but both points remain within the same high-accuracy region, demonstrating the robustness of the surface structure.
> >
> >
> >
> > We hope this response addresses your concerns and clarifies the unique contributions of our work. We welcome any further discussions to explore these distinctions in greater depth.

---

> > > ### Comment · Reviewer_LRVF · 2024-12-01
> > > **Response**
> > >
> > > Thank you for your response and in particular for patiently explaining the differences between your method and the related works.  I am more convinced now regareding the novelty of the outlined approach! I have increased my score accordingly.

---

> > > > ### Author Response · Authors · 2024-12-01
> > > > **Appreciation for Your Feedback**
> > > >
> > > > Thank you again for your thoughtful feedback and for taking the time to review our response! We're delighted to hear that our explanations were helpful. We would be happy to engage in further discussions or address any additional questions you may have.

---

### Meta-Review · Area_Chair_JP5R · 2024-12-18

**Metareview:**

The paper extends mode connectivity in neural network loss surfaces to two-dimensional Bezier surfaces.  Specifically, the authors extend the observations and methodology of [1] from training 1-dimensional Bezier curves to 2-dimensional Bezier surfaces. They first define a loss function for training the surface with 4 fixed corner points. Then, they develop a method for optimizing the loss where they first fit the edges of the surface and then fit the inner part. The authors show that the proposed method finds surfaces with low train loss and high test accuracy across multiple architectures and image classification datasets. They also provide results on ensembling of points within the surface and model merging.

Strengths:
- The paper is well-written
- The proposed methodology is sound
- The method works well and achieves the goal set by the authors

Weaknesses:
- The paper is a pretty direct extension of the observations and methodology in [1]. Also, [2] has previously demonstrated that generally mode connectivity holds with high-dimensional connecting manifolds. So the observations and insights are not completely novel. The authors argue that [2] is qualitatively different as it constructs a locally-linear surface; however, the full surface is still non-linear, and it's not clear why a smooth Bezier surface is a major improvement over a simplicial complex.
- The experiments are conducted on smaller datasets, same as [1]. It is worth noting that [1] was written in 2018, and the standard for empirical studies should be higher now. Having experiments at ImageNet scale would be good.
- It would be nice to see results of merging models trained on different tasks, e.g. different subsets of data, where the merging is actually beneficial. The current model merging results are more of a proof-of-concept and not a practical improvement.

Decision recommendation: Despite the limitations, the paper provides new results on mode connectivity, the methodology is sound and the presentation is strong. I recommend accepting the paper, but I also suggest that the authors should add more discussion of differences with [2] and the other relevant papers highlighted by reviewers in the final version of the paper.

[1] Loss Surfaces, Mode Connectivity, and Fast Ensembling of DNNs
Timur Garipov, Pavel Izmailov, Dmitrii Podoprikhin, Dmitry P. Vetrov, Andrew G. Wilson

[2] Loss Surface Simplexes for Mode Connecting Volumes and Fast Ensembling
Gregory Benton, Wesley Maddox, Sanae Lotfi, Andrew Gordon Gordon Wilson

**Additional Comments On Reviewer Discussion:**

The reviewers unanimously recommend accepting the paper. Reviewers engaged with the rebuttal from the authors, and 2 of them raised their scores based on the rebuttal. The reviewers had concerns about the relationship to prior work, as well as requested additional details on the method and results. The reviewers were satisfied with the responses from the authors.

---

### Decision · Program_Chairs · 2025-01-22

Accept (Poster)